# Predictive learning as a network mechanism for extracting low-dimensional latent space representations

Stefano Recanatesi [1✉], Matthew Farrell[2], Guillaume Lajoie [3,4], Sophie Deneve[5], Mattia Rigotti [6,8] & Eric Shea-Brown[1,2,7,8]

Artificial neural networks have recently achieved many successes in solving sequential processing and planning tasks. Their success is often ascribed to the emergence of the task's low-dimensional latent structure in the network activity – i.e., in the learned neural representations. Here, we investigate the hypothesis that a means for generating representations with easily accessed low-dimensional latent structure, possibly reflecting an underlying semantic organization, is through learning to predict observations about the world. Specifically, we ask whether and when network mechanisms for sensory prediction coincide with those for extracting the underlying latent variables. Using a recurrent neural network model trained to predict a sequence of observations we show that network dynamics exhibit low-dimensional but nonlinearly transformed representations of sensory inputs that map the latent structure of the sensory environment. We quantify these results using nonlinear measures of intrinsic dimensionality and linear decodability of latent variables, and provide mathematical arguments for why such useful predictive representations emerge. We focus throughout on how our results can aid the analysis and interpretation of experimental data.

[1] University of Washington Center for Computational Neuroscience and Swartz Center for Theoretical Neuroscience, Seattle, WA, USA. [2] Department of Applied Mathematics, University of Washington, Seattle, WA, USA. [3] Department of Mathematics and Statistics, Université de Montréal, Montreal, QC, Canada. [4] Mila-Quebec Artificial Intelligence Institute, Montreal, QC, Canada. [5] Group for Neural Theory, Ecole Normal Superieur, Paris, France. [6] IBM Research AI, Yorktown Heights, NY, USA. [7] Allen Institute for Brain Science, Seattle, WA, USA. [8] These authors contributed equally: Mattia Rigotti, Eric Shea-Brown. ✉email: stefanor@uw.edu

Neural network representations are often described as encoding latent information from a corpus of data[1–7]. Similarly, the brain forms representations to help it overcome a formidable challenge: to organize episodes, tasks, and behavior according to a priori unknown latent variables underlying the experienced sensory information. In this paper, motivated by the literature suggesting that these efficient representations are instrumental for the brain's ability to solve a variety of tasks[8–10], we ask: How does such an organization of information emerge?

In the context of artificial neural networks, two related bodies of work have shown that this can occur due to the process of prediction—giving rise to predictive representations. First, neural networks are able to extract latent semantic characteristics from linguistic corpora when trained to predict the context in which a given word appears[11–14]. The resulting neural representations of words (known as word embeddings) have emergent geometric properties that reflect the semantic meaning of the words they represent[15]. Second, models learning to encode for future sensory information give rise to internal representations that encode task-related maps useful for goal-directed behavior[9,16–18].

As predictive mechanisms have been conjectured to be implemented across distinct neural circuits[19–21], characterizing predictive representations can then shed light on where and how the brain exploits such mechanisms to organize sensory information. Our goal is to build theoretical and data-analytic tools that explain why a predictive learning process leads to low-dimensional maps of the latent structure of the underlying tasks —and what the general features of such maps in neural recordings might be. This links predictive learning in neural networks with existing mechanisms of extracting latent structure[22–24] and low-dimensional representations from data[25].

We begin with an introductory example of how predictive learning enables the extraction of latent variables characterizing the regularity of transitions among a set of discrete "states", each of which generates a different observation about the world. Then we focus on a model where observations are generated from continuous latent variables embedded in a low-dimensional manifold. We focus on the special case of spatial exploration, in which the latent variables are the position and orientation of an agent in the spatial environment, and the observations are high-dimensional sensory inputs specific to a given position and orientation. The predictive learning task we study is to predict future observations. Our central question is whether a recurrent neural network (RNN) trained on this predictive learning task will extract representations of the underlying low-dimensional latent variables.

We develop analytical tools to reveal the low-dimensional structure of representations created by predictive learning. Crucial to this is the distinction between linear[26–30] and non-linear dimensionality[31,32], which allows us to uncover what we call latent space signal transfer, wherein latent variables become increasingly linearly decodable from the top principal components of the neural representation as learning progresses. Latent space signal transfer is accompanied by clear trends in the linear and nonlinear dimensionality of the underlying representation manifold, and potentially gives rise to the the formation of neurons with localized activations on the nonlinear manifold, manifold cells[33]. Importantly, while each of these phenomena could separately find its origin in a mechanism different from predictive learning, they altogether provide a strong measurable feature of predictive learning that expect further testing in both neural and machine-learning experiments. We conclude by extending our framework to the analysis of both neural data and a second task—arm-reaching movements.

## Results

**Predictive learning and latent representations: a simple example**. In predictive learning a neural network learns to minimize the errors between its output at the present time and a stream of future observations. This is a predictive framework in the temporal domain, where the prediction is along the time axis[20]. At each time $t$ an agent observes the state of a system $o_t$ and takes an action $a_t$ out of a set of possible actions. The agent is prompted to learn that, given $(o_t, a_t)$, it will next observe $o_{t+1}$.

We begin by illustrating our core idea— that predictive learning leads neural networks to represent the latent spaces underlying their inputs—in a simple setting. We study the task shown in Fig. 1a, where the state of the system is in one of $N_s = 25$ states. To each state is associated a unique set of five random cards that the agent observes whenever it is in that state. The states are organized on a two-dimensional lattice—the latent space. Observations have no dependence on the lattice structure, as they are randomly assigned to each state with statistics that are completely independent from one state to the next. On the other hand, actions are defined on the lattice: at each time $t$ the agent either randomly moves to one out of the four neighboring states by selecting the corresponding action or remains in the same state. Movements, when they occur are thus along the four cardinal directions N, S, W, E used to indicate the corresponding action. Meanwhile, 0 denotes the action corresponding to no movement, for a total of $N_a = 5$ possible actions.

The agent solves this predictive task when, prompted with a pair $(o_t, a_t)$, it correctly predicts the upcoming observation $o_{t+1}$. A priori, this task does not require the agent to extract information about the underlying lattice structure of the state space. Indeed the agent could solve the task with at least two possible strategies: (1) by associating with each observation (set of cards) the next observation via a collection of $N_s \times N_a$ distinct relationships $(o_t, a_t) \mapsto o_{t+1}$ (combinatorial solution), or (2) via a simple set of relationships that exploit the underlying lattice structure of the state space. In this second scenario the agent would uncover the lattice structure while using it to map actions to predictions. This solution thus presupposes an internal representation of the latent space and we refer to it as predictive representation solution. The critical difference between the combinatorial and predictive representation solutions is that the latter extracts a representation of the latent space while the former doesnot, cfr. Fig. 1b.

We train a simple two-layer network on this card-game task: to predict the future observation given inputs of the current observation and action, Fig. 1c. We focus on the first layer that receives the joint input of actions and observations. In this example observations are encoded with a one-hot representation, formally turning the problem into a classification task. Upon learning, by means of Stochastic Gradient Descent (SGD), the network develops an internal representation in the hidden layer for each of the 125 input pairs $(o_t, a_t)$.

Visualizing these internal representations in the space of principal components of neural activations, the underlying latent structure of the state space appears (Fig. 1d.) This lattice-like structure is a joint representation of observations and actions. This representation emerges over the course of learning: initially, the representation of each observation-action pair $(o_t, a_t)$ does not reflect the underlying latent space, see Fig. 1d. The development of the latent space representation can be clearly visualized across stages of the learning process (see Fig. S1).

Additionally, if we remove the actions from the input to the network but still training it to perform prediction, the network still learns a representation that partially reflects the latent space, Fig. 1e, though this time it is distorted (cf. Fig. S2).

Below, we will demonstrate this phenomenon in other more complex settings, but we first pause to build intuition for why it

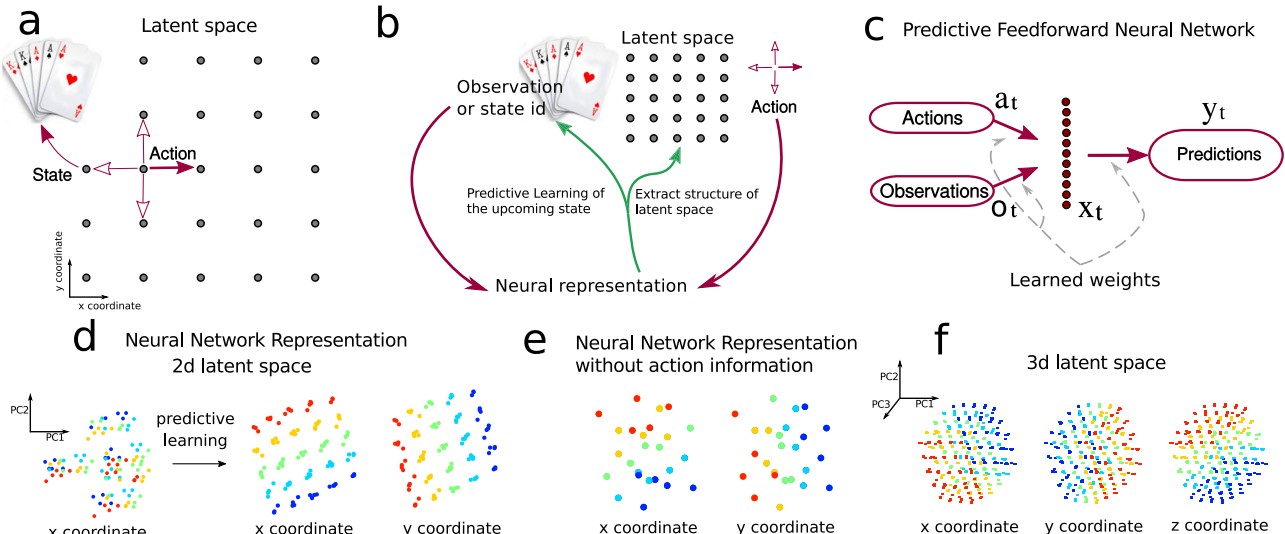

**Fig. 1 Predictive network solving a card-game task. a** Description of the latent space underlying the task. **b** Illustration of the task and information flow diagram: the neural representation receives state observations and actions and extracts the latent space structure by means of predicting upcoming observations. **c** Diagram of the network's structure. The diagram highlights the layer studied here, although the network has a two layers, where the second layer serves as a decoder. **d** The network's neural representation: activity in the hidden layer plotted vs. principal components PCs 1 and 2 of hidden layer activity. For each observation-action pair ($o_t$, $a_t$), the corresponding activation is colored by the position of the state that the network predicts: x-coordinate (left plot, before and after learning) and y-coordinate (right plot). **e** Same as panel **d** in the absence of the action as a input to the network. **f** Same as panel **d** for a three-dimensional latent space.

occurs within neural networks. We start by noticing that upon learning the five actions $a \in \{N, S, W, E, \theta\}$ are mapped to a fixed vector $\boldsymbol{w}_a$, which is added to the state representation $\boldsymbol{w}_s$ every time the corresponding action is selected:

$$\boldsymbol{x}_{s,a} = \tanh(\boldsymbol{w}_s + \boldsymbol{w}_a + \boldsymbol{b}), \tag{1}$$

where $\boldsymbol{b}$ is a learned bias parameter. Specifically, consider the representation $\boldsymbol{x}$ in the network for predicting a state $s'$ located immediately above (to the N) of the state $s$, in two scenarios. In the first, $s'$ is arrived at from $s$, after the action $a = N$. This gives the representation

$$\boldsymbol{x}_{s,N} = \tanh(\boldsymbol{w}_s + \boldsymbol{w}_N + \boldsymbol{b}) \tag{2}$$

In the second, $s'$ is arrived at from $s'$, after the null action:

$$\boldsymbol{x}_{s',0} = \tanh(\boldsymbol{w}' + \boldsymbol{w}_0 + \boldsymbol{b}) \tag{3}$$

Both of these activations must be read out to return the same prediction: $s'$. While this could occur in principle if the readout operation learned to collapse different representations to the same readout, the network learns a simpler solution in which the representations (Eq. (2)) and (Eq. (3)) are equal (cf.[34,35]), so that $\boldsymbol{x}_{s',0} = \boldsymbol{x}_{s,N}$ implies:

$$\boldsymbol{w}_{s'} - \boldsymbol{w}_s = \boldsymbol{w}_N - \boldsymbol{w}_0 \tag{4}$$

for any pair of states $s, s'$ linked by the action $a = N$. This implies that (up to the hyperbolic tangent non-linearity), the representation of the states is acted upon by the action in a translational invariant way in the direction of the action $\boldsymbol{w}_N - \boldsymbol{w}_0$. This is true for any of the actions $N, S, W, E$, and for any starting state $s$. Thus, the representation inherits an approximate translation invariance—the characteristic property of a lattice structure. This invariance confers a geometrical structure upon the learned neural representation that reflects the latent space. This phenomenon directly generalizes to lattices of higher dimension, as shown in Fig. 1f.

We note that this analysis holds precisely when the learning process enforces representations of the same decoded state to be nearly identical—which occurs in all of our simulations and is

predicted by other numerical and theoretical studies[34,35]—and holds approximately when it tends to cluster these together. By contrast, in a general combinatorial solution of Eq. (4) each observation action pair could be linked to the upcoming state independently, $\boldsymbol{x}_{s,N} \neq \boldsymbol{x}_{s',0}$.

We can apply related ideas to begin to understand more challenging case in which the prediction task is performed without knowledge of the action, so that only observations are passed as input to the network. As we showed in figure (Fig. 1e) above, in this case the internal representation still partially reflects the latent space. This is not because the set of observations as a whole carries any information about the latent space, but because the effect of the actions—to bind nearby states together—is reflected in the statistics of the sequence of observations. Thus, through making predictions about future observations, the network still learns to bind states that occur nearby in time together, extracting the latent space (cf. Suppl. Mat. Sec. 2.4).

We next generalize the predictive learning framework to two different, more complex benchmark tasks of neuroscientific interest: spatial exploration and arm-reaching movements.

**Predictive learning extracts latent space representations in a spatial exploration task**. We focus on predictive learning in a spatial exploration task in order to generalize the previous example to show how predictive learning extracts the low-dimensional latent structure from a high-dimensional sensory stream (Fig. 2a) and to introduce novel metrics, which quantify such process.

In the spatial exploration task an agent traverses a square open arena. Traversing the environment, the actions taken determine a trajectory in three spaces: the latent space, which defines the agent's (or animal's) state in the environment, the observation space of the agent's sensory experience, and the neural activation space of its neural representation. We introduce the task defining these three spaces.

The latent space, similarly to the card-game example, is the set of spatial coordinates that identifies the agent's state, $(x, y, \theta)$,

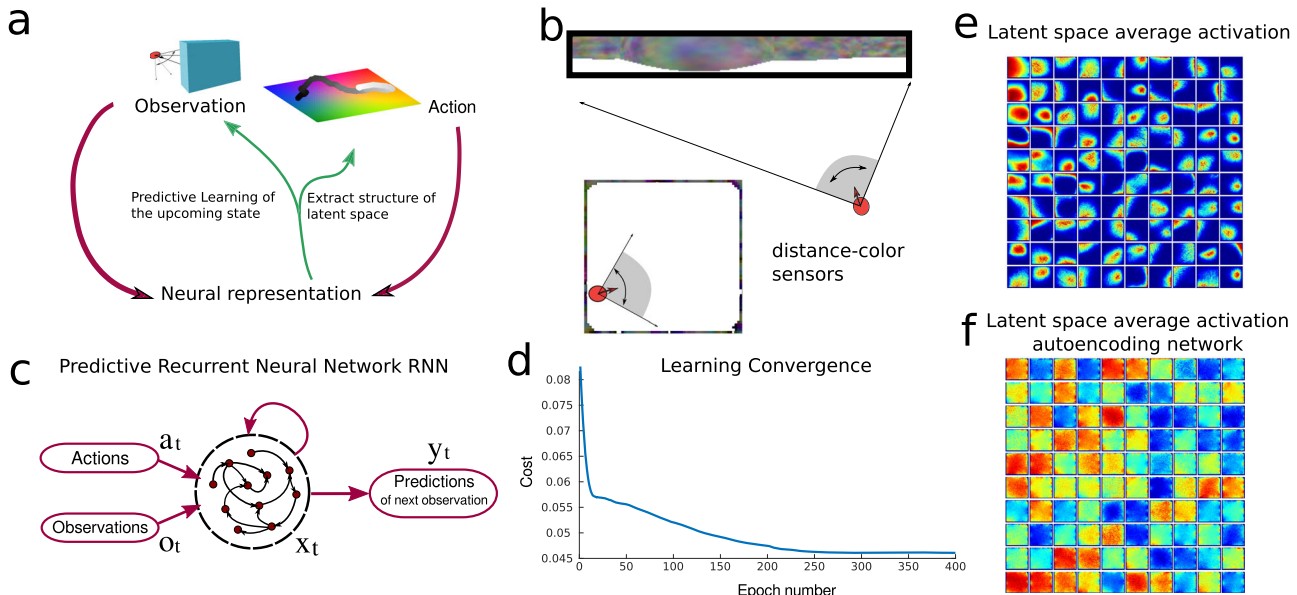

**Fig. 2 Predictive network solving an exploration task. a** Information flow diagram of the task: an agent explores a two-dimensional environment (latent space) through actions and receives observations regarding it. The network's task is to predict the next sensory observation. By learning to do so it recovers information regarding the underlying hidden latent space. **b** Illustration of the agent with sensors in the square environment where the walls have been colored (cfr. Methods). The sensors span a 90º degree angle and register the color and distance of the wall along their respective directions. **c** Diagram of the predictive recurrent neural network: the network receives actions and observations as inputs and is trained to output the upcoming sensory observation. **d** Cost during training for the network (cf. Methods). **e** Average activity of 100 neurons (each of the 100 neurons average activity is showed in one of the small 100 quadrants) against the $x$, $y$ coordinates of the environment, showing place-related activity. **f** Same as panel **e** for a RNN trained to autoencode its input observations.

where $x$ and $y$ are position and $\theta$ is its direction. The observation space is defined in terms of the agent's ability to sense the surrounding environment. To model this we consider the case where the agent senses both visual and distal information from the environment's walls—the agent is equipped with sensors that span a 90º visual cone centered on its current direction $\theta$ reporting distance and color of the environment's wall along their directions, Fig. 2b. The environment the agent navigates is a discrete grid of $64 \times 64$ locations. Each wall tile, one at each wall location, is first colored randomly and then a narrow spatial autocorrelation is applied, see Fig. 2b. The number of sensors $N_s$ is chosen so that observations across sensors are independent $N_s = 5$.

We consider the case where the agent's actions are correlated in time but do not depend on the observations—random exploration. At each step the agent's direction $\theta$ is updated by a small random angle $d\theta$ drawn from a Gaussian distribution centered at zero and with a variance of 30º. The agent then moves to the discrete grid location most aligned with the updated direction $\theta + d\theta$ (unless it is occupied by a wall; cfr. Methods for details). Actions are performed by the agent with respect to its allocentric framework, so that there are nine possible choices: for each location there are eight neighboring ones plus the possibility of remaining in the same location. While the agent moves in the environment it collects a stream of observations.

In predictive learning, the agent learns to predict the upcoming sensory observation, Fig. 2c. It achieves this by minimizing the difference between its prediction $y_t$ at time $t$ and the upcoming observation $o_{t+1}$: $C = \Sigma_t \|y_t - o_{t+1}\|^2$, Fig. 2d. We refer to the activations of the units of the trained RNN as its internal predictive representation. The RNN can be thought as a model of the agent's brain area carrying out the task. As the agent learns to predict the next observation, its representation is influenced both by the observation space (since the task is defined purely in terms of observations) and by the latent space (since the actions are

defined on it); a priori, it is not obvious which space's influence will be stronger. In this example, we used a more general recurrent network rather than the simplest-possible feedforward setup in the first example of Fig. 1; this allows information from the stream of sensory observations to be integrated over time, a feature especially important in more challenging settings when instantaneous sensory information may be only partially informative of the current state.

A first indication that, by the end of learning, neurons encode the latent space is given by the fact that individual neurons develop spatial tuning Fig. 2e. The neural representation has extracted information about the latent space from the observations, without any explicit prompt to do so. In the Suppl. Mat. (Figs. S7–S12), we show how this phenomenon is robust to alterations of the sensory observations and network architecture.

However, when the same network learns, based on the same input sequence, to reproduce the current observation (autoencoding framework corresponding to a cost $C = \Sigma_t \|y_t - o_t\|^2$) rather than predict the upcoming one, individual neurons do not appear to develop spatial tuning, Fig. 2f and Suppl. Mat. (Figs. S10–11).

**Metrics for predictive learning and latent representations.** How —and to what extent—does the neural population as a whole represent the latent space? This question demands quantitative answers. To this end we develop novel methods for analyzing neural representation manifolds, and three metrics that capture the dynamical and geometrical properties of the representation manifold. These are predictive error, latent signal transfer and dimensionality gain. While the first of these is specific to predictive frameworks, the other two could be interpreted as general metrics to quantify the process of extraction of a low-dimensional latent space from data. Below we illustrate these metrics in the context of the spatial exploration task (cf. Figs. S3–5 for a detailed analysis and more examples of such metrics).

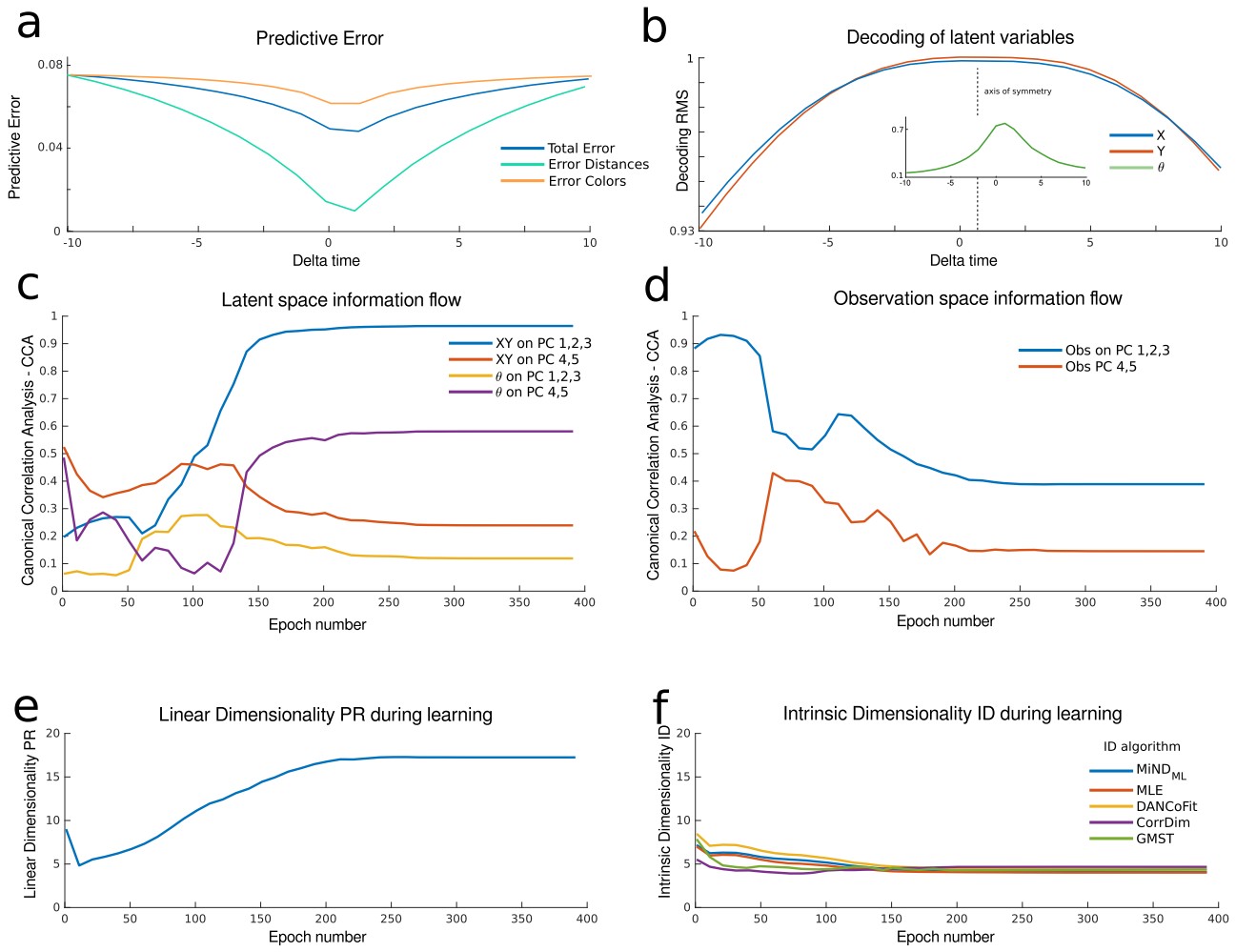

**Fig. 3 Learning the predictive representation. a** Predictive error ($L_2$ norm) in blue between the network's output and the observation as a function of the lag (Delta $t$). In red average $L_2$ norm between the observation at time 0 and at a lag Delta $t$. **b** Linear decoding of latent variables. RMS measure of the linear decoding of $(x, y, \theta)$ at time Delta $t$ from the neural representation at time 0. The dotted line highlights the axis of symmetry of the curves. **c** Signal transfer analysis: Canonical Correlation Analysis between PCs of the neural representation and the latent space. The lines correspond to the average of the canonical correlations between the highlighted variables. **d** Same as panel **c** but for the observation space. **e** Participation ratio of the representation during learning. **f** Intrinsic dimensionality (ID) of the representation during learning. Five different intrinsic dimensionality estimators are used (cfr. Methods).

**Predictive error**. The network's task is to predict future observations. Owing to correlations in the sensory input itself from one timestep to the next, to verify that the network is actually making predictions we first ask whether the network's output is most similar to the upcoming observation rather than current or previous ones[36]. This can be captured by the absolute difference between the current output of the network and the stream of observations at any time, which we refer to as predictive error. If this is skewed towards the upcoming observation (see Fig. 3a blue line), it suggests that the network predicts elements of upcoming observations. This measure relies on knowledge of the network's output and of the stream of observations. An allied measure of this effect relies on the ability to decode past vs. future latent states from the current neural representation. If the decoding error is skewed for future (vs. past) latent states, this also suggests that the network predicts future states. Figure 3b shows that this is the case for the spatial exploration network: it codes for future latent variables as well as current and past ones, with the axis of symmetry for decoding the spatial coordinates $x$, $y$ located close to the future value $\Delta t = 1$ (cf. Fig. S13 for a comparison with neural data). Similarly the axis of symmetry for the angle $\theta$ is located closer to $\Delta t = 1$, although in this case the analysis is

confounded by the fact that actions carry partial information regarding $\theta$.

**Latent signal transfer**. We next introduce a feature of predictive learning that tracks how the neural representation reflects the latent space over the course of learning. This quantifies the phenomenon visible by eye in the introductory example of Fig. 1d. To define the latent signal transfer metric, at each stage of learning we compute the average of the canonical correlation (CC) coefficients between the representation projected into its PCs, and latent space variables $x, y, \theta$. The blue line in Fig. 3c shows the average of the CC coefficients between the representation in PCs 1 to 3 and the position $x, y$ of the agent in latent space. When the average CC coefficient is 1, all the signal regarding $x, y$ has been transferred onto PCs 1 to 3 in a linear fashion. A similar interpretation holds for the other curves: in sum, they track the formation of explicit representations of latent variables that are accessible via linear decoding .

Figure 3c shows that, between epoch 50 and 150, most of the information regarding the latent space moves onto the first few PC modes of the neural representation. The same analysis can be

carried out with respect to observation space variables. This is shown in Fig. 3d, where the decreasing trend indicates that the observation space signal flows out of the first few PC components as learning progresses. Altogether Fig. 3c, d show that the representation, as interpreted through PC components, encodes more information about the latent space as opposed to the observation space as learning progresses (blue and red lines).

**Dimensionality gain**. Finally, motivated by the fact that the latent spaces of interest are lower-dimensional, we introduce metrics that allow us to quantify the extent to which the learned neural representations have a similar dimension.

We begin by noting that the latent signal transfer analysis (Fig. 3c, d) suggests that predictive learning might have formed a low-D neural representation. However, when we measure the dimensionality of the neural representation with a linear dimensionality metric, the participation ratio (PR), we observe that dimensionality actually increases over the course of learning Fig. 3e. Instead, measuring the dimensionality of the neural representaion with nonlinear techniques sensitive to the local curvature of the representation manifold—yielding the intrinsic dimensionality (ID)—shows that the dimensionality rather than increasing at most decreases through learning.

This dichotomy can be interpreted by means of two different demands that shape network representations. On one hand, the representation is prompted to encode high-dimensional observations; on the other, it extracts the regularity of a low-dimensional latent space. While the high dimensionality of the observations is a global property, referring to the collection of many observations, the regularity of the latent space is induced on a local scale, as neural representations relate to their possible neighbors via the action. These demands lead the linear dimensionality PR, measuring a global property of the representation manifold, and the nonlinear dimensionality ID, measuring more local properties, to have opposite trends. This interpretation is supported by further experiments and the next example we study, that arm-reaching movements, in which the network is prompted to predict a lower-dimensional observation signal. To encapsulate this phenomenon we suggest the metric of dimensionality gain (DG), which is the ratio between the linear global dimensionality and the nonlinear local dimensionality of the representation manifold. Higher values of DG thus capture the network's ability to extract a low-dimensional representation of a high-dimensional stream of observations. In the example of Fig. 3e, f, DG ≈ 3.5 upon learning.

**The role of prediction in extracting latent representations**. To show how the three metrics just described characterize predictive learning, we compare representations learned in the same networks but without the demand for prediction (as in Fig. 2f). In Fig. 4 we show how predictive error, latent signal transfer and dimensionality properties of the network differ in these two cases. The comparison is carried out by training 50 different networks of smaller size (100 neurons) on either the predictive task or a non-predictive version in which the network outputs observations received on the current timestep. In sum, comparing each of the metrics introduced above for the predictive vs. non-protective cases shows that, while predictive learning extracts a low-dimensional manifold encoding for the latent variables, non-predictive learning in these networks does not.

One point here bears further discussion. While Fig. 4e shows that ID is lower in the predictive vs. non-predictive case, this may seem surprising because there are grounds to expect that ID would be equal in these cases. These grounds are that the observations are produced as a map from a low-dimensional

latent space in both cases, so that if the network directly encodes them, it should admit a similar low-dimensional parametrization and hence similar ID in both cases as well. The resolution comes from the fact that ID, despite being a local measure, is based on statistical properties of points sampled from a manifold (cf. "Methods"). So if the manifold appears higher dimensional, despite having a parametrization, which is low-dimensional, then ID would point to a higher dimension. In other terms ID is sensitive to the manifold's smoothness and can be taken as a measure of it for manifolds parameterized by a fixed number of variables. This problem is known in the literature as multiscaling and different ID measures are more or less robust to it[31].

Finally we note that, in Suppl. Mat. Figs. S7–12, we describe a series of 12 other control networks that show how results on the role of prediction are robust against a number of factors such as noise. These results show that predictive models outperform non-predictive models in the encoding of latent variables, at least when such encoding is probed by means of linear measures (cf. Fig. S7).

**Visualizing the structure of learned neural population manifolds: signal transfer and neural manifold cells**. The metrics just introduced capture properties of the neural representation at the population level via useful numbers that can be plotted over the course of learning. Here, we pause to visualize the underlying population representations in two complementary ways.

The first visualization is directly related to the metric of latent space signal transfer. In Fig. 5a the neural representation projected into the space of its first three PCs, colored according to each of the three latent variables $x$, $y$, and $\theta$. Each point in these plots corresponds to the neural representation at a specific moment in time, and the color of the point is determined by the position or orientation of the agent in the latent environment at that moment. This shows visually that, after learning, the agent's location $x$, $y$ is systematically encoded in the first three PCs, while PCs four and five encode the agent's orientation $\theta$, Fig. 5b. This corresponds to the high values of latent space signal transfer seen at the end of learning in Fig. 3c. We next turn to visualize whether the observation variables are similarly encoded in the network representation. Figure 5c shows that, while the first three PCs do encode distance, they do not appear to encode the sensor-averaged color in any of the three RGB (red, green, blue) channels. Intriguingly, this is a consequence of learning: average color information is encoded in the first PCs in the beginning of learning as suggested by the signal transfer measure (cfr. Fig. 3c), but less in the end of it. Taken together, the visualizations in Fig. 5a, b support the conclusion from the signal transfer metrics that the network allocates most of its internal variability to the encoding of latent variables.

These visualizations of population level neural coding, as well as plots of single neuron tuning as in Fig. 2e, require foreign knowledge of the latent space variables. However, in many settings, neither the values or nature of these variables maybe known in advance. How can we proceed in these cases? We now introduce a second strategy for visualizing neural activity, via an emerging concept that we refer to as neural manifold cells[33,37].

Figure 5d shows the activity of the same 100 neurons in Fig. 2e averaged over "locations" in the space spanned by the first two PCs of the neural population activity itself. This shows tuning of individual neurons, but not with respect to motor, stimulus, or environmental variables as is typically studied—but rather with respect to population level neural activity. The approach reveals a similarity between the well known phenomenon of place cells tuned to a location in the environment and neural manifold cells tuned to a "location" on the principal components of their neural population manifold (we make this relationship made more

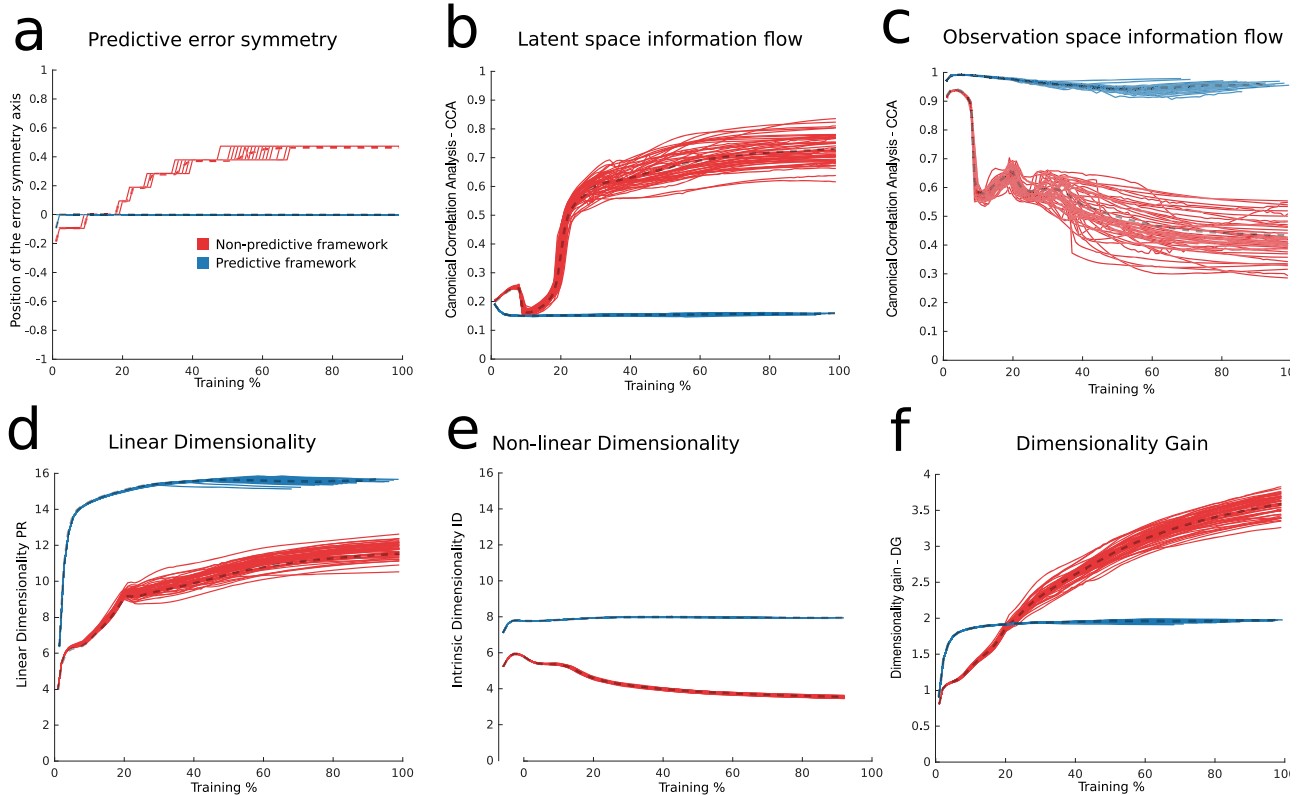

**Fig. 4 Comparison between predictive and non-predictive learning.** We train 50 networks of 100 neurons in each of the predictive and non-predictive conditions and equalize the learning axis between the two to highlight the trends of the different measures. **a** Predictive error. The position of the predictive error symmetry axis plotted throughout learning for the predictive and non-predictive network ensembles. The symmetry axis position is the one that minimizes a L2 norm between the predictive error curve (cf. Fig. 3a) and its reflection through the symmetry axis. **b** Latent signal transfer analysis. A canonical correlation analysis is performed between the latent space and the top PCs of the neural representation at every epoch, and the average of the two canonical correlations (for coordinates x and y) is shown. **c** Observation signal transfer analysis. The canonical correlation analysis, same as panel **b**, is performed between the top PCs of the observations and the top PCs of the network's representation. **d** Linear dimensionality (PR) throughout learning. **e** Non-linear dimensionality (ID) throughout learning. **f** Dimensionality gain (DG) throughout learning.

explicit in the context of hippocampal data in Fig. S13). Overall, this shows that receptive fields localized not just in the latent, but also in the principle component, spaces can arise naturally through predictive learning.

**Predictive learning extracts latent representations of arm-reaching movements**. While the spatial exploration task studied above is a useful proving ground, given the clear role played by latent spatial variables, we wished to illustrate the broader scope of the effects of predictive learning. Thus, we next apply this framework to a different task, that of predicting arm-reaching movements. We model arm movements as a dynamical system with forward and inverse kinematics according to the mitrovic model[38,39]. In this model, movements in the 2d sagittal plane of the upper right limb are modeled as a function of six muscles, Fig. 6a. The muscles control, by means of dynamical equations, two angles: the angle in between the upperarm and the line of the shoulders, and the angle in between the forearm and upperarm. The position of the elbow and wrist is then a nonlinear trigonometric function of these angles and of the lengths of the upperarm and forearm.

We cast this system into predictive learning by generating randomly correlated binary input pulses, which signal the contraction of one of the six muscles through the forward kinematics equations, resulting in exploratory movements of the arm.

We train the predictive recurrent network to predict future (x,y) locations of both the elbow and the wrist given their current locations and the input to the six muscles. This replicates the

spatial exploration task description in terms of observations and actions, where observations are in this case thought to be the current locations of the elbow and wrist with respect to the shoulder Fig. 6b and actions are muscular contraction signals.

Upon learning, the network successfully predicts future observations and extracts in its neural representation the values of the underlying latent variables that ultimately regulate the movements: the two angles, see Fig. 6c, e. Owing to the low dimensionality of the observations compared with the spatial exploration task, and the fact that they are partially colinear with the latent variables, latent space signal transfer increases over the course of learning as before, but observation space signal transfer does not decay.

For the same reason, the linear dimensionality (PR), as it increases through learning, achieves a lower final value. The latent variable extraction is accompanied by the localization of neural activations on the neural population manifold and on the latent space as shown in Fig. 6f, g replicating the results shown for the spatial exploration task. Furthermore, we analyzed neural recordings in the primary motor cortex[40,41] during a motor task, as an example of how our analysis of representations of arm-reaching movements could inform future data analyses and experiments, cf. Fig. S14.

**Network mechanisms that create low-D representations through prediction**. In our introductory example of the card-game, we gave some mathematical reasoning for how simple

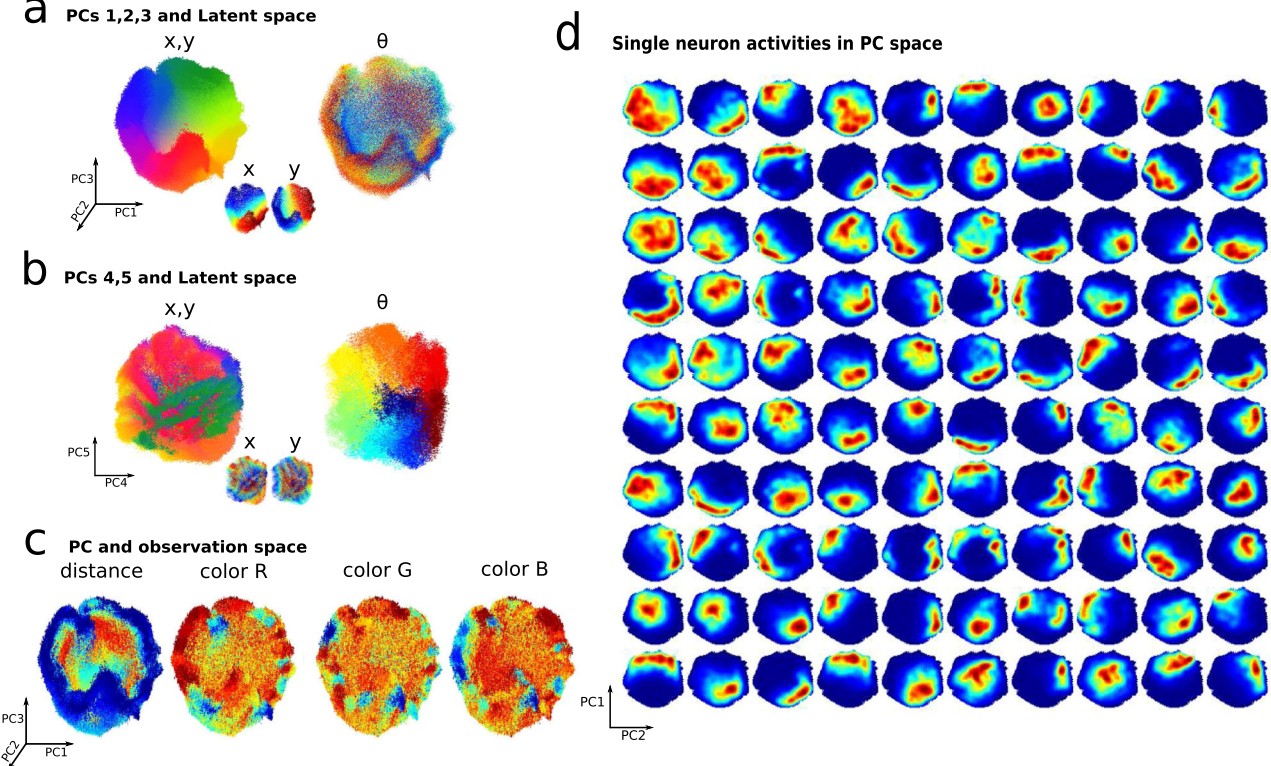

**Fig. 5 Features of the learned predictive representation. a** 100,000 points of the neural network representation, corresponding to an equal number of steps for the agent's exploration, are shown projected into the space spanned by PCs 1 to 3 of the learned representation, and colored, respectively, according to x, y latent variables (cfr. Fig. 1a for color code) and θ. **b** Same as panel **b** but for PCs 4 and 5. **c** Same as panel **a** but colored with respect to the mean distance or color activations of the agent's sensors. In this specific example, the first five PC components explain, respectively, 13.7%, 11.4%, 10.2%, 5.5%, 5.4% of the total neural variance. **d** Manifold cell activations: average activity of 100 neurons on the manifold (here displayed for the first PCs 1 and 2.). The activity of each neuron (one per quadrant) is averaged as the population activity is in a specific "location" on the neural manifold in the space spanned by PCs 1 and 2.

feedforward networks trained to predict their future inputs (observations) can extract the structure of the latent space underlying those observations. Here, we formalize this idea and extend it to recurrent networks, as used for the more general spatial and motor exploration settings studied above. Here, the RNN is governed by the equations:

$$r_t = g\big(Wr_{t-1} + W_o o_t + W_a a_t\big)$$
$$y_t = g\big(W_{out} r_t\big) \qquad (5)$$

where $W, W_o, W_a, W_{out}$ are the weight matrices and $y$ is the output exploited to minimize the predictive cost $\mathcal{C}_{pred} = \sum_t |y_t - o_{t+1}|^2$. Following the same logic as for the card-game task, we consider two independent network updates, denoted by A and B respectively, which lead up to the same observation $o_{t+1}$, read out from identical representations $r_t^A = r_t^B$. Again, up to nonlinear corrections, this gives the condition:

$$r_{t-1}^A - r_{t-1}^B = W^{-1}\big(W_o(o_t^A - o_t^B) + W_a(a_t^A - a_t^B)\big) \qquad (6)$$

which is an analogous to Eq. (4). From here, we consider two different scenarios.

In the first, the action term dominates. This gives an identical case to the one already analyzed in the introductory section Eq. (4): the action acts on the neural representation in a translationally invariant way. As before, this results in representations corresponding to different observations being translated with respect to one another similarly to how the action translates among them in the underlying latent space. For the spatial exploration task this corresponds to the product of a two-dimensional lattice and a circle

(angle); for the arm-reaching task this corresponds to the product of two angles.

In the second scenario, the observation term dominates. Observations at the current time define a set of possible observations at the next timestep, those related to the current observation via one of the possible actions from the current point in the latent space. Extending the reasoning above suggests that representations $r_A$ and $r_B$ of latent states $A$ and $B$ should be similar according to the overlap in this set of possible next-timestep observations. This again suggests that the structure of latent space will be inherited by representations, as it is only states that are related by one action that can map to the same next-timestep observation. This is indeed what we find: Fig. 1e and Figs. S7–10 (case without actions) show how the latent space emerges in neural representations in predictive networks even in the absence of action inputs. However, the Supplemental Sec. S1.2 does show that these representations carry latent information in a less regular way when actions are not provided to the networks.

Taken together, these results show that the network's representation is shaped by the latent space by means of learning to predict future inputs. This connects to novel approached that have recently led to important progress in the theory of deep learning[42–44] by applying group theory to analyze neural networks[45,46]. Through this emerging perspective predictive networks, when prompted with the current observation of the state of a system ($o$) can be analyzed as if they were asked to output the transformed observation upon applying the action of a group element $g_a$: $o \mapsto g_a(o)$. In our setup we use the generators of the group instead of all possible group elements. As the network

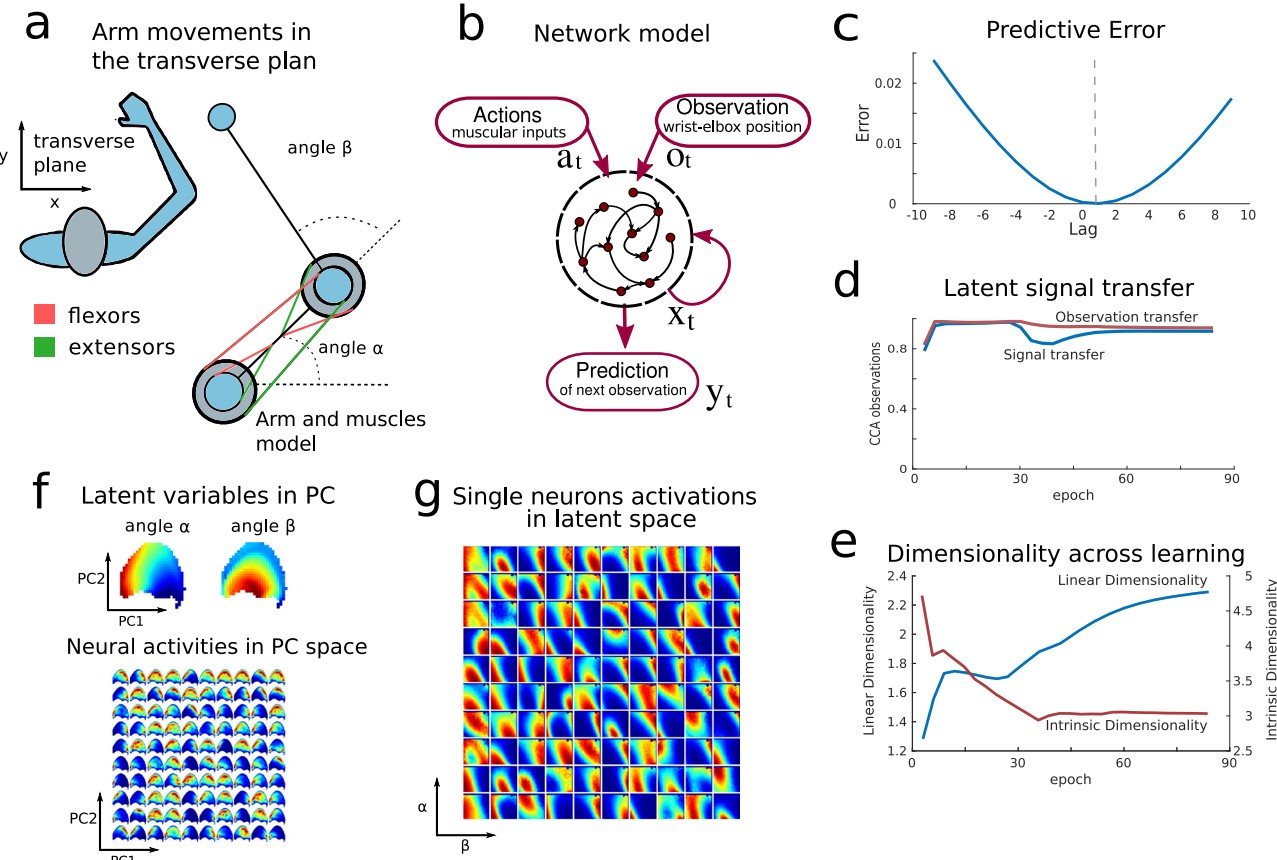

**Fig. 6 Predictive representations of arm-reaching movements. a** Plane transverse to the dynamic of arm-reaching movements. The muscle model is shown and the two latent angular variables $\alpha$ and $\beta$. **b** Recurrent network model. **c** Predictive error upon training. The symmetry axis is around lag +1 indicating that the network is carrying out the prediction correctly. **d** Latent signal transfer and observation signal transfer. **e** Dimensionality trends across learning for both linear (PR) and nonlinear (ID) dimensionality measures. **f** Top: principal components space (PCs 1–2) colored by the average angles $\alpha$, $\beta$ for each location. Bottom: average activity of neurons in the space spanned by the top 2 PCs. Each subplot represents the average activity of a single neurons. Neurons are ranked according to their average firing rates. The most active neuron is in the top left corner, the second in the first column second row and so on for all the neurons. **g** Average activity of neurons in latent space $\alpha$, $\beta$. Each subplot corresponds to the neuron in panel **f**.

learns to apply group actions $g_a$ to its representation, it transforms, through its layers, the given observation $o$ into a neural representation onto which the action acts as a group element.

At this stage the network's representation inherits the geometry of what is called the group's representation. For example, in the spatial exploration example, the states in which the agent can be found are defined by the Special Euclidean group of rotations and translations in two dimensions SE(2). In our framework the actions of the agent correspond to the group generators for translations—reflecting minimal translational movements of the agent (the angle, corresponding to the rotation degree, is not directly provided). Thus, the action passed to the network is formally the one relative to the translation subgroup, and it is provided in vectorial form. As these group generators act as vectorial translations on the neural representations, a definite geometry is inherited by the network representation: the translation subgroup of SE(2) is encoded as a two-dimensional lattice[47]. This is a more general way to arrive at the conclusions of the direct calculations taken above.

The analysis above shows how the structure of the latent space shapes the structure of neural representations. This structure can be clearly visualized in many of the plots presented above. Moreover, it is reflected in the metrics we introduce in at least two ways. First, we expect that states being represented in a transitionally invariant way will lead to the ability to decode

states from neural representations; how this plays out for the principal components of neural activity that are used for plotting neural activity above and for the metric of latent space signal transfer is described using results from the linear algebra of Toeplitz matrices in Supplemental Secs. S2.1–2.3. Second, states being represented in a transitionally invariant way leads to an approximate parameterization of neural activity via terms of the latent space, corresponding to the lower values of intrinsic dimensionality also measured above.

By contrast, as an autoencoder does not compute the action of a group element on its input, is not generally expected to build a representation with structure induced by that group. Nonetheless a group theoretic approach to autoencoders still enables insights into why autoencoders develop activations reminiscent of receptive fields[48]. In the Suppl. Mat. Sec. 2.5 we provide further considerations on the locality of receptive fields mainly inspired by ref. [37].

## Discussion

How the brain extracts information about the latent structures of the external world, given only its sensory observations, is a long-standing question. Here, we show that the computation of predicting future inputs can contribute to this process, giving rise to to low-dimensional neural representation of the underlying latent spaces in artificial neural networks. We demonstrate this

phenomenon in a sequence of gradually more complex simulations and by providing basic mathematical arguments that indicate its generality.

What features of neural responses, or representations, characterize predictive learning? When the observations to be predicted arise from an environment with an underlying low-dimensional latent structure, e.g., in the case of spatial exploration or arm-reaching movements, our work suggests several distinct features. First, the predictive error shows that neural representations are biased towards encoding upcoming observations or latent variables. Second the latent structure underlying the observations is transferred onto the representation progressively through learning (Latent Signal Transfer, cf. Fig. 5). Finally, the dimensionality of the set of neural responses will likely appear high when assessed with standard linear measures, such as participation ratio[28,29]. However, when assessed through nonlinear metrics sensitive to the dimensionality of curved manifolds, the dimensionality will be lower, in the ideal case tending to the number of independent latent variables.

This last feature is the result of neural responses being strongly tuned to the variables, which parameterize the neural representation manifold (cfr. Fig. 5d). An established example of such strong coding is the locality of neural receptive fields in latent space (e.g., place fields). Here, we observe an allied phenomenon, that of manifold cells with local receptive fields on the manifold of population-wide neural responses. This is a feature that can be explored in artificial network studies of complex data, or in experimental settings (cf. proof-of-concept data analysis in Suppl. Mat. Fig. S13) where the underlying latent variables do not need to be known in advance. This feature connects to recent work on understanding neuronal representations through the lens of dimensionality[27–29,37,49]. Overall, these features provide a quantitative framework to compare representations across conditions that can be applied both in machine learning (e.g., to compare learning schemes and overall mechanisms of extracting latent signals from data) and in brain circuits (e.g., to compare coding in distinct brain areas).

Our findings should not be taken as a theory of a specific brain area but rather as a formulation of a general connection between predictive coding and the extraction of latent information from sensory data. For example, our model falls short in explaining mechanistically key elements of spatial maps individuated in hippocampal recordings, such as the emergence of place cells and their relation to direction or grid cells. However, it does suggest that predictive learning is a mechanism that enables the binding of sensory information beyond spatial exploration and towards the more general notion of semantically related episodes. While traditionally distinct theories of hippocampus involve declarative memory[50]) and spatial exploration[51], considerable effort has been devoted to reconciling these apparently contrasting views[52–55]. In particular, Eichenbaum[54] proposed that the hippocampus supports a semantic relational network that organizes related episodes to subserve sequential planning[8,9,56]. Here, we posit that prediction—with its ability to extract latent information—may serve as such a mechanism to generate semantic relational networks. In particular, we speculate that relevant semantic relations are encoded by neural representations of low intrinsic dimensionality, which are constructed by predictive learning to reflect the relevant latent variables in a task. Our results substantiate and build on the importance of allied frameworks in constructing such relational networks[15,16,57]. Overall the predictive learning framework provides a potential alternative of generating hippocampal representations, which differs from both attractor[58,59] and path-integration models[60,61], while maintaining elements of both these models. Discerning the underlying differences and similarities will require careful future investigations.

From an algorithmic and computational perspective, our proposal is motivated by the recent success of predictive models in machine-learning tasks that require vector representations reflecting semantic relationships in the data. Information retrieval and computational linguistics have benefited enormously from the geometric properties of word embeddings learned by predictive models[11–13,62]. Furthermore, prediction over observations has been used as an auxiliary task in reinforcement learning to acquire representations favoring goal-directed learning[9,16–18]. Alongside these studies there are other emerging frameworks that are related to the predictive learning networks we analyze: contrastive predictive coding[63,64], information theoretic approaches[65,66] and world models[67]. Furthermore, our contribution shall also be seen in light of computational models studying neurons with optic flow selectivity[68,69].

Predictive learning is a general framework that goes beyond the examples analyzed here, and future work can expand in other directions (text, visual processing, behavioral tasks, etc.) that may open new theoretical advances and new implications for learning and generalization. It will also be exciting to adapt and test these ideas for the analysis of large-scale population recordings of in vivo neural data—ideally longitudinally, so that the evolution of learned neural representations can be tracked with metrics such as the emergence of a low-D neural representation manifold, predictive error, latent signal transfer and dimensionality gain. A very interesting possibility is that this might uncover the presence of latent variables in tasks where they were previously unsuspected or unidentified. Our techniques require no advance knowledge of the latent variables. The consequence is that both the number and identity of latent variables can be discovered by analysis of a learned neural response manifold, as studied in other settings[62,70–72].

Furthermore, it will be important to develop a formal connection between predictive learning mechanisms[73,74] and reinforcement learning (RL) paradigms[9,75] in both model-free and model-based schemes[76–78]. This has the potential to build a general framework that could uncover predictive learning behavior in both animals and humans. One step here would be to extend existing RL paradigms to scenarios where making predictions is important even in the absence of rewards[79–82].

## Methods

**Card-game network**. We generate a two-dimensional 5 x 5 grid of states, which is the latent space. To each state, we randomly assign a random set of five cards from a deck of 40, sampled with no repetition. This serves as an example of observations associated to states, which are fully random, independent, and of arbitrary complexity. In particular the dimensionality of the observation is not tied to the dimensionality of the latent space. We generate $10^6$ state transitions following the five actions as defined in the main text. Upon generating such sequence of states we train a feedforward network to predict upcoming obeservations given current ones. The network is a two-layer network with 100 neurons in both layers, the first with sigmoidal transfer function and the second with hyperbolic tangent followed by a binary cross-entropy cost function. Both actions and observations have a one-hot encoding. All weights are initialized with random normal matrices. Training is performed on 80% of the sequence and validated on the remaining 20% utilizing a RMSprop optimizer (parameters: learning constant = 0.0001, $\alpha = 0.95$, $\epsilon$ regularizer = $1 \cdot 10^{-7}$). The learning rate was reduced of a factor 0.5 if the validation loss did not decrease for eight consecutive epochs (reducing on plateau scheme). Training was stopped after 25 epochs with no improvement in the validation loss (min delta of variation 5e-5). The neural network used for Fig. 2e is identical to the one just described, except that the output is read out at the second layer (the hyperbolic tangent layer) with mean-squared error. This is to account for the fact that the prediction, when actions are not passed to the network, is probabilistic towards neighboring states. All simulations were performed in Keras.

**Neural network model for the spatial exploration task**. We study a recurrent neural network (RNN) that generates predictive neural representations during the exploration of partially observable environments. RNNs are suited to processing sequence-to-sequence tasks[83] and the state of a recurrent network is a function of the history of previous inputs and can thus be exploited to learn contextually appropriate responses to a new given input[84–86].

Figure 2c illustrates the RNN model: at a given time $t$ the RNN receives as input an observation vector $\overrightarrow{o}$ and a vector representation of the action $\overrightarrow{a}$. The internal state $\overrightarrow{r}^t$ of the network is updated and used to generate the network's output through the following set of equations:

$$r_t = g(W r_{t-1} + W_o o_t + W_a a_t)$$
$$y_t = g(W_{\text{out}} r_t) \tag{7}$$

The RNN is trained to predict the observation at the next timestep by minimizing the first cost function, or alternatively to autoencode its input, via the predictive and non-predictive cost functions, respectively:

$$\mathcal{C}_{\text{pred}} = \frac{1}{T} \sum_{t=0}^{T-1} ||o_{t+1} - y_t||^2 ,$$
$$\mathcal{C}_{\text{non-pred}} = \frac{1}{T} \sum_{t=0}^{T-1} ||o_t - y_t||^2 . \tag{8}$$

Networks were trained by minimizing the cost function in Eq. (8) via backpropagation through time[87]. While RNNs are known to be difficult to train in many cases[88], a simple vanilla RNN model with hyperbolic tangent activation function was able to learn our task, Fig. 2d.

The connectivity matrix of the recurrent network was initialized to the identity[89,90], while input and output connectivity matrices were initialized to be random matrices. Individual weights were sampled from a normal distribution with mean zero and standard deviation 0.02. The network had 500 recurrent units (with the exception noted below), while the input and output size depended on the task as defined by the environment. Each epoch of training corresponded to $T = 10^6$ time steps.

All other training details were the same as reported for the card-game example. For the simulations of Fig. 5, we trained 100 networks of 100 neurons: 50 networks in the predictive case and 50 networks in the non-predictive case (cf. Eq. (8) with equal instantiation of the rest of parameters).

**Description of the spatial environment**. modeled the spatial exploration task in two dimensions. We simulated the exploration of the agent in a square maze tessellated by a grid of evenly spaced cells ($64 \times 64 = 4096$ locations). At every time $t$ the agent was in a given location in the maze and headed in a direction $\varphi \in [0, 2\pi)$. The agent executed a random walk in the maze, which was simulated as follows. At every step in the simulation an action was selected by updating the direction variable $\theta$ stochastically with $d\theta$ (i.i.d. sampled from a Gaussian distribution with variance $\sigma_{\text{theta}}^2 = 0.5$ rad). The agent then attempted a move to the cell, among the eight adjacent ones, that was best aligned to $\theta$. The move occurred unless the target cell was occupied by a wall, in which case the agent remained in the current position but updated its angle with an increment twice the size of a regular one: $\sigma_{\text{theta}}^2 = 1.0$ rad. To ensure coherence between updates in the direction $\theta$ and the cell towards which the agent just moved, we required each update in $d\theta$ to be towards the direction of the agent's last movement $d_a$ so that $d\theta \cdot (\theta - d_a)$ would always be positive, where $d_a$ assumed one of 8 values depending on the action taken by the agent.

The chosen action was encoded in a one-hot vector that indexed the movement. The actions were discrete choices $a_t \in [0..8]$ correlating with the head direction but distinct from it. This was indeed a continuous variable $\theta_t \in [0, 2\pi)$. Moreover, knowledge of the action didn't provide direct information about the agent's direction and observation; in other words, there was no direct correspondance between the action taken and the observation collected as for each location and action there were many possible directions the agent could point towards and consequently as many possible observations.

As the agent explored the environment it collected, through a set of $N_s = 5$ sensors, observations of the distance and color of the walls along five different directions equally spaced in a 90 degree visual cone centered at $\varphi$. Thus it recorded, for each sensor, four variables at every timestep: the distance from the wall and the RGB components of the color of the wall. This information was represented by a vector $o_t$ of size $5 \times 4 = 20$. Such a vector, together with the action represented as a one-hot representation, was fed as input into the network and used for the training procedure. The walls were initially colored so that each tile corresponding to a wall carried a random color (i.e., three uniformly randomly generated numbers in the interval [0,1]). A Gaussian filter of variance two tiles was then used, for each color channel, to make the color representations smooth. Figure 2b shows an example of such an environment.

**Predictive error**. The predictive error is a direct generalization of Eq. (8) as a function of a time lag variable:

$$\mathcal{C}_{\text{pred}}(\text{lag}) = \frac{1}{T} \sum_{t=0}^{T-1} ||o_{t+\text{lag}} - y_t||^2, \tag{9}$$

so that it is possible to verify that the output of the network $y$ is most similar, on average, to the upcoming observation rather than the current observation.

**Latent signal transfer**. The latent signal transfer measure was obtained by performing a canonical correlation analysis (CCA) between two spaces: the top 3 PC components of the network's representation and other variables as specified in the text, e.g., latent variables (x,y). CCA extracts the directions of maximal correlation between the two spaces returning a set of canonical correlations. Latent signal transfer is then taken to be the average of these canonical correlations, which are as many as the minimum between the ranks of the two spaces.

**Nonlinear dimensionality: intrinsic dimensionality**. While research on estimating intrinsic dimensionality (ID) is advancing, there is still no single decisive algorithm to do so; rather, we adopt the recommended practice of computing and reporting several (here, five) different estimates of ID based on distinct ideas[31,32]. The set of techniques we use include: MiND$_{ML}$[91], MLE[92], DancoFit[93], CorrDim[94], and GMST[95,96]. These techniques follow the selection criteria illustrated in ref. [31], emphasizing the ability to handle high-dimensional data (in our case hundreds of dimensions) and being robust, efficient, and reliable; we refer the reader to ref. [25] for a useful comparison. We implement these techniques using the code from the the authors available online[31,92,93], "out of the box" without modifying hyperparameters.

A simple intuition regarding for some of the selected techniques builds on the notion of correlation dimension, which derives from the following idea. Consider a manifold $\mathcal{M}$ of dimensionality $d$ embedded in $\text{IR}^N$ and a set of points uniformly sampled from the manifold. For each point build a ball of radius $r$ (denoted as $B_r$), then the number of points within $B_r$ (denoted as $\#B_r$) can be analyzed as a function of $r$ and be found to scale as $\#B_r \sim r^d$ at least for small $r$. This scaling can be exploited to estimate $d$.

**Description of arm-reaching movements model**. To model arm-reaching movements we used a kinematic model of the arm muscles[97,98]. The arm kinematics were modeled in the transverse plane by analyzing the effect of six muscles on the arm dynamics, cf. Fig. 6a. The activation signals for the muscles were used as actions in our model. For each of the six muscles, we used a pulsed binary signal where at each instant in time the pulse can be turned on or off. These activation signals are filtered and passed to the equations of inverse kinematics of the muscles, which regulate muscular contraction. Such muscle dynamics drives the arm dynamics according to the Mitrovic model[38,99,100]. All the details regarding the implementations of this model can be found on the Github repository we adopted for the simulations https://github.com/jeremiedecock/pyarm and in the code we provide. The most relevant feature of this model for our study is the fact that the six-dimensional muscle activity drives nonlinear dynamics in the two-dimensional latent space described by the two angles $\alpha$, $\beta$ in Fig. 6a.

**Reporting summary**. Further information on research design is available in the Nature Research Reporting Summary linked to this article.

## Data availability
All data generated through the simulations generated is made available from the corresponding author upon reasonable request.

## Code availability
All code is made available from the corresponding author upon reasonable request.

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

## Acknowledgements

E.S.B. is supported by NSF Grant 1514743, wishes to thank the Allen Institute for Brain Science founders, Paul and Jody Allen, for their vision, encouragement, and support. G.L. is supported by an NSERC Discovery Grant (RGPIN-2018-04821) and the FRQNT Young Investigator Startup Program (2019NC253251). Part of this work was conducted during an internship at IBM Research. The authors would like to acknowledge the numerous colleagues who have helped generate the ideas of the paper. In particular, we thank Luca Mazzucato (University of Oregon, USA), Kameron Decker Harris (University of Washington, USA), Stefan Mihalas (Allen Institute for Brain Science, USA), Greg Wayne (DeepMind, UK), and Alon Rubin (Weizmann Institute, Israel), Cengiz Pehlevan (Harvard University).

## Author contributions

S.R.: conceptualization, formal analysis, software, validation, visualization, writing. M.F.: conceptualization, formal analysis, review and editing. G.L.: conceptualization, formal analysis, review and editing. S.D.: conceptualization, formal analysis, review and editing. M.R.: conceptualization, project administration, supervision, writing, review and editing. E.S.B.: conceptualization, project administration, supervision, writing, review and editing.

## Competing interests

The authors declare no competing interests.
