## [Peer Review File · Nature Communications]

Reviewers' Comments:

Reviewer #1:

Remarks to the Author:

Predictive learning extracts latent space representations from sensory observations

Reviewed by David Sussillo

Contributions:

The primary contribution of this work is to analyze with both RNNs and with some mathematics how predictive learning with an agent in a 2D environment gives rise to a low-dimensional latent representation of that environment, by learning $\{\text{observation}_t, \text{action}_t\} \rightarrow \text{observation}_{t+1}$, . The work compares both the linear embedding of this representation and the nonlinear intrinsic dimensionality, and shows how, through learning, this representation becomes more pronounced, in the sense of occupying larger amounts of variance explained, as measured by signal location in the top PCs.

To my knowledge, the work is novel, interesting and technically correct. However, I admit, I have no idea what to make of it. My confusion stems from a number of sources:

Comparisons to hippocampus (is this work meant to be about what the hippocampus does?). I acknowledge I am not an expert in this domain.

My sense that the article is making possibly overly broad claims that predictive learning is THE thing required for low-d representations.

My wondering what the recurrent nature of the RNN is bringing to the result.

My strong opinion that the manuscript requires substantial effort in order to improve its readability to the broad audience it hopes to reach.

Major concerns:

1. What is this paper really about and how does it relate to neuroscience?

I read the position piece [22], highlighted in both the manuscript Introduction and Discussion. Its central tenet appears to be that, "hippocampus contributes by supporting memory necessary for successful navigation rather than by performing navigational computations per se" and "it is difficult to empirically associate the hippocampus with spatial computations per se, as opposed to memory for the spatial parameters and events relevant to ongoing behavior in space". The position appears to be primarily concerned with knocking down the idea that hippocampus function is solely related to integration of self motion cues, and rather is part of a relational processing system that includes navigation.

Indeed, in the manuscript discussion, the authors state that "Finally we note that the responses are reminiscent of the types of place-related activity observed in the hippocampus and entorhinal cortex, lending in particular mechanistic grounding to the recent proposal by (22) that the hippocampus builds a semantic relational network. We argue that relevant semantic relations are encoded by neural representation of low intrinsic dimensionality, and in turn these are being constructed by predictive learning to reflect the relevant latent variables in a task." What does this mean? Please expand this point further with examples. Did you create a semantic relational network? Can you point to it and to examples, with your concrete mechanistic model?

In what way is this manuscript useful to experimentalists (a useful paper need not be, I am just trying

to help clarify and understand)? Is the idea that they can measure DG and then infer there is a low-dimensional representation based on predictive learning?

2a. The statements are too strong given the results. Reading this paper, one would conclude that predictive learning is THE means by which low-dimensional latent structure is created in neural networks. However, there are a large number of examples of (low-dimensional) meaningful latent structure being learned in artificial networks. The authors point out word embeddings, for example. Please take the time to explain what is special about predictive learning.

"This latent space signal transfer is another signature of predictive learning that we can exploit and track through training." Does this have to do strictly with predictive learning? Would another kind of non-predictive task also do this? More generally, the single task that is studied somewhere during the manuscript gets converted to the phrase "predictive learning". This feels like a far too general switch, as only a single task is studied here, aside from a few related controls.

An additional example of an overly strong conclusion is that predictive learning is the way to learn low-D latent structure because auto-encoding networks do not. One may merely conclude that auto-encoding is not great for learning low-dimensional latent structure. Going further, what is the motivation for the non-predictive case, e.g. $|o_{\{t\}} - y_{\{t\}}|^2$? Again, the language here feels a bit strong, in that there are many ways in which a task involves stimuli but is not predictive in nature. Perhaps "Non-predictive learning fails to extract low-D latent manifold" should be converted to "Auto-encoding RNNs fail to learn ..." E.g. what happens if you were to ask the network to predict distances and wall colors just outside of the set of angles that are typically measured, but from the same time step (o_t)? This task would not involve prediction through time but might still learn an interesting representation of the space.

Another example: "Here, we investigate the hypothesis that representations with low-dimensional latent structure, reflecting such semantic organization, result from learning to predict observations about the world."

I would be much more comfortable with: "Here, we investigate the hypothesis that *a means for generating* representations with low-dimensional latent structure, reflecting such semantic organization, result from learning to predict observations about the world."

2b. I am not certain I believe the generality of the results. "Our techniques require no advance knowledge of what the latent variables are, or even how many of them there are. The consequence is that both the number and identity of latent variables can be discovered by analysis of a learned neural response manifold, as studied in other settings by (43, 46–48). We introduce latent signal transfer as a viable way to uncover the relevant variables fig. 3d: as the response manifold is learned, the position of population responses along the manifold can be increasingly well predicted by the true low-dimensional latent variables, but increasingly poorly predicted by irrelevant variables."

Can you show this on an example that isn't so precisely setup? E.g. is there another, unrelated example you can try this on? Or are these very general and strong statements leaning on the mathematical section? Please explain.

3. What utility is given by the recurrence? In what way is the hidden state of the RNN being used? I.e. what are the dynamics? Stated yet another way, what would have been different or broken had the model been a feed-forward network? This is mentioned in the controls section, but basically asserted that feed-forward networks "hinder the development of predictive representations with the key properties described above". Given the mathematical analysis in the "A neural network mechanism for low-D representation manifolds through predictive learning" section, it seems like understanding this is of central importance to the measure of the paper. For example, could not the linearization

techniques around fixed points for RNNs be used to study this (Sussillo & Barak, 2013)? Does the RNN make a big plane attractor? Understanding how the dynamics and representations interact seems pretty important, and likely accessible, and would help validate the mathematics derived in the manuscript.

I am guessing, though I am uncertain, that the latent space signal transfer is related to the dynamics required to perform the task. Given the control of an auto-encoder being able to learn, but not replicating the main dimensionality results, this seems plausible.

4. The organization of the paper is very confusing.

The jump from figure 1 being about an RNN and task setup, to figure 2 being about a mathematical analysis, back to figure 3 being about the RNN really confused me. I needed more guidance through the exposition. Just to confirm, is Figure 2 unrelated to the RNN trained on predicting next-step sensory input? If so, it was not clear to me until I got to the first paragraph of the "The learned neural representation manifold" section. E.g. Figure 1 has an RNN in it, it seems pretty clear that Figure 2 is going to be about that.

How do you know the latent variables are a generative model for the observations? This phrasing confused me significantly. There is a traditional generative model $p(x|z)$ in this manuscript?

Moderate concerns:

Fig 1a is your go-to figure for reader orientation. Respectfully, I think it is failing in that regard and needs work. E.g. I could not tell what part of 1a is the agent, where my eye should begin to look in the panel, and to where my eye should saccade next.

I think more needs to be explained in the exposition about the nonlinear dimensionality methods.

Other researchers have been working on RNN models that give rise to grid cells. It seems relevant to cite them (perhaps I missed it) even if the focus may be somewhat different.

Cueva & Wei, ICLR 2018

<https://arxiv.org/pdf/1803.07770.pdf>

They find grid cells by converting from head direction and speed and predicting x,y position. Is this a task that would or would not give rise to the phenomena that you study in this manuscript?

Technical questions:

If the latent space wasn't low-dimensional, what would happen with a predictive learning network?

Is there noise in the network activations? I ask because it seems significant to the authors that the latent variables are encoded in the top PCs. However, I do not see why this really matters if there is no noise, as long as it can be read out correctly.

Should I make anything out of x,y going to pc 123 and theta going pcs 4,5? Typically, signals are mixed in the top PCs simply because top PCs explain highest variance and are otherwise unrelated to the task.

"The latter necessarily implies a low dimensional representation, the same as latent space." - Why is this necessary? Can it not be the case that the $W_{r_{t-1}}$ is high-dimensional but the readout W_{out}

reads out neural state necessary? I.e. r can wander in directions orthogonal to the readout, thus potentially being high-dimensional, while not affecting performance.

Can you decode the latent variables at all in the auto-encoding networks? If so, to what extent?

Minor concerns:

There is a goodly amount of jargon in this paper. I encourage the authors to internally justify each term and see if they cannot reduce to a minimal set and thereby reduce the cognitive load involved in reading this piece somewhat.

Fig 1e caption: I do not know why the place cell is "x,y coordinates of the latent space", isn't it the x, y coordinates of where the agent is?

Fig 1e,f caption: "(one per small quadrant)", I think some information is missing.

Fig 1f: Is that meant to be a space denoting the head direction angle? Please improve clarity

Fig 2e caption: "PR dependence on the size of the Gaussian field", yet the figure shows DG on on the y axis

"A neural network mechanism for low-D representation ..."

Discussion of policies and "off policy actions" are a bit out of nowhere. While the methods are typically relegated to the end, it seems important to include the bits necessary during the main exposition so as not to confuse the reader.

Given that you train the RNN on random trajectories, it is probably worth spelling out what this means for the agent's dynamical system $F(x)$, and what you would expect to learn in the RNN as a result.

Perhaps explain semantic relational network in the discussion?

Canonical correlation analysis vs canonical covariation analysis

In discussion, did you mean Fig 4d?: "uncover the relevant variables fig. 3d: as"

Reviewer #2:

Remarks to the Author:

[Contributions]

This work explores the underlying mechanism of recurrent predictive learning methods in extracting semantic latent information. First, it provides approximate mathematical arguments, and second, it shows experimental evidence by studying the representations of a one-layer RNN model in a simplified random walk environment.

[Strengths]

1) Originality: Exploring the profound reasons for the effectiveness of predictive learning is a natural extension to this field, but to my knowledge has not been done before.

2) Quality: The qualitative and quantitative correlation analyses between the latent variables and the recurrent representations clearly demonstrate the advantage of predictive learning over non-predictive learning (auto-encoders).

3) Significance: Some analytical techniques in this paper about the neural representations, e.g. the latent signal transfer experiment in Fig. 3, might be of interest to practitioners who want to design

new predictive learning methods.

[Improvements]

4) Clarity: This paper is not organized very clearly. I encourage the authors to use a paragraph to highlight the main contributions and the layout of this paper.

5) Methodology: The authors introduce a new metric, the Dimensionality Gain (DG). However, it is not clear to me what the motivation of this new metric is. Is it the higher the better (though in Fig. 5f, predictive learning has a greater DG value, I cannot find significant difference between Fig. 5d and Fig. 5e)? Can I use it to testify the effectiveness of a new predictive learning method? I think the authors could further clarify its significance with more evidence or at least more qualitative explanations to the result.

6) Experiments: All empirical results are based on a very simplified task. I have some concerns with this so-called navigation task.

- Why did the authors make the agent take random walks instead of learning a navigation policy. With random walk, there are less dynamical coherences lying beneath the sequential observations. The process of predicting future observations might rely more on the actions.

- The environment is too simple: the latent location variables (x, y) are discrete, the observations are merely 20d vectors. How about using a more complicated environment, like DeepMind Lab, with continuous latent variables and RGB frame observations.

- The sequential action inputs might reveal the latent variable, theta, which makes the correlation analysis regarding theta and neural representations less convincing.

- In such an easy environment, future observations are easy to predict given the very first observation and a trajectory of actions. As the training epoch grows, the neural network might have remembered these information and make prediction without any glimpse of any new observations. Thus, I have to cast doubt on the correlation analysis between the observations and the neural representation PCs.

- Last but not least, the authors use an extremely simplified network, a one-layer RNN. I understand for mathematical analysis, simplification and approximation is acceptable. But in the experimental analysis, the authors might investigate some more frequently-used network structures, such as LSTM and GRU, which obviously violate the linear assumption in Eq. (7) and (8).

7) References: The authors may cite some recent references in spatiotemporal data forecasting and model-based planning, e.g. World Model, which have successfully applied predictive learning to its downstream tasks.

8) Typos: right column of page 2: it should be Fig. 1c and Fig. 1d.

[Conclusion]

I believe this paper is tackling an important problem, but makes some claims without strong evidence (see comment 6). I thus cannot recommend acceptance at the current time but might improve my score if enough evidence is provided.

Reviewer #3:

Remarks to the Author:

Majors

The manuscript does not attempt to tie the theoretical simulations and arguments back to data or established models of data (e.g. place cells, head direction cells, grid cells, boundary cells etc). I recognize that it attempts to make a connection between the latent space representation to resemble place cells but falls short of a detailed analysis and linking that to reported data.

The work overall and sections in detail (see examples in minors) requires better motivation. Currently,

it reads: Let's take the latest training methods (RNN) and simulate a predictive task and study the latent representations in principle component space. It's unclear to me why this approach is novel or relevant to the audience of this journal.

It's unclear to me how the current finding would provide a tool to analyze existing or new data in a new light. Maybe, it would help if the authors propose such a tool and take a few data sets (on place cells, head direction cells, grid cells) that have been collected in navigation tasks and apply their machinery there. E.g. the Moser lab provided data here <https://www.ntnu.edu/kavli/research/grid-cell-data>.

The task of predicting the environmental layout for the next step is rather arbitrary and constructed. It is in the line of work of auto-encoders (here with some predictive time dynamics) that exists for a long time. Tasks for navigation – if this is what the authors are aiming for; which citations to work of memory representations in rats seems to suggest -- are semi-supervised in nature. The current proposal has a conceptual mismatch here or implicitly assumes that such predictive auto-encoders are a pre-requisite to successful navigation.

The methods section and theoretical arguments provide the reader with some insight into the framework but the manuscript falls short in providing a reproducible description of the model. I've also given the supplemental material a quick read and could not find a complete description there either. I acknowledge that the authors have provided source code but I suggest that this does not replace a description of the model that would allow the reader to reproduce their findings.

Minors (details)

Page 2: "demonstrate our main result: that the network uncovers the low-dimensional latent space structure in the course of optimizing its future predictions.." The approach is not novel (e.g. Beardsley & Viana, Computational modeling of optic flow selectivity in MSTd, Comput Neural Sys 1998; study latent representation – hidden layer – in non-recurrent network to identify representations related to optic flow patterns for a navigation task, or Olshausen & Field. Emergence of simple-cell receptive field properties by learning a sparse code for natural images. Nature 1996; study an auto-encoder for images with a sparseness constraint where representations show Gabor filter-like shapes).

Page 2: "The agent we consider is equipped with simple sensors that span a visual cone of 90o centered on its current direction _." Why where these values chosen? Why do we have color? Does it matter? Why not use odors or whiskers as sensory input? Please be reasonable and motivate well. I'm unsure how reasonable is the RGB color space for rats; if you indeed try to simulate rat vision.

Page 2: "Actions are performed by the agent with respect to its allocentric framework, so that there are nine possible choices" What's transforming the sensory signals into allocentric coordinates?

Page 2: "In predictive learning, the RNN learns to predict the upcoming sensory observation..." Which species has a mechanism like that: To predict the shape/layout after each single step? Mental rotation tasks of 3d objects might come closest to this. But humans are poor at this task.

Page 3: "How does the neural population as a whole represent the latent space?" Why does this questions matter? Please motivate.

Page 3: "We can view the tuning curve of a single neuron (Fig. 2b) on the response manifold to obtain the manifold tuning curve of this neuron (Fig. 2d). In the next section we will analyze in more depth." The projection is neighborhood preserving when visualized in a 3D PC1-3 space. Do you see this as a result of visualizing in PCs or as part of the mechanism or as part of the task being fairly continuous?

Page 3: "The fact that the representation manifold has two dimensions is revealed by a measure

known as Intrinsic Dimensionality (ID), whose formal definition relies on concepts of Riemannian geometry for smooth manifolds" A motivation for why we need to do this analysis is missing. What's the intuition behind this analysis?

Page 4: "This shows that the agent's location x,y is systematically encoded in the first three PCs, while PCs four and five encode the agent's orientation , Fig. 3b." Interesting. Do you think this is a result of choosing PC1-5 in that way? There seems to be some form of neighboring preserving smoothness here; but this time it is split across modalities.

Page 8: "We find that predictive learning in recurrent neural networks (RNNs) leads to an intriguing answer, as it automatically constructs a low dimensional neural representation of the latent space." Here predictive seems to be the key. Could you show the same holds true for other tasks, e.g. for a grasping, visual tracking, ...?

Page 9: "Finally we note that the responses are reminiscent of the types of place-related activity observed in the hippocampus and entorhinal cortex, lending in particular mechanistic grounding to the recent proposal..." Can you quantify the place cells more strictly, e.g. by fitting Gaussian models and reporting r^2 or by defining a place cell score?

Reviewer #4:

Remarks to the Author:

The manuscript addresses a set of key questions in theoretical neuroscience, related to emergence of useful neural representations as a result of unsupervised learning and adaptation to the sensory environment. Specifically the authors study whether recurrent networks trained via predictive learning to predict their future sensory inputs, will learn to uncover and represent the (typically low-dimensional)

latent variables underlying their sensory inputs. Addressing such questions is very welcome, and the current study could potentially be very impactful and influential.

However, I find several major shortcomings in the current manuscript. My two major criticisms are:

- I found some of the main claims and proposed hypotheses of the paper not to be clearly stated or sufficiently elaborated

to distinguish them from trivial (or in some cases tautological) interpretations. In other cases I found the evidence provided for the claims to be incomplete.

- The mathematical analysis provided to explain some of the findings was quite lacking in rigor.

I understand that a fully rigorous analysis may be difficult or impossible. But at least highly non-rigorous steps of the arguments

could/should e.g. be corroborated by clearly presented directly relevant empirical evidence from the simulations.

Below I will go into the above (and other more minor comments) in more detail.

But first, I'll summarize the three main claims of the paper (all summarized at the end of Introduction):

Predictive Learning (PL) by an recurrent neural net (RNN) creates or leads to:

C1. low-dimensional neural representations, reflecting the few latent variables (LVs) underlying sensory observations.

C2. what they call Latent Space Signal Transfer (LSST).

C3. localized receptive fields (RFs).

Moreover C1-3 are taken as "signatures" of PL.

1. Regarding C1: Be more precise in stating this hypothesis and potentially modify it. In particular, when you say "low-dimensional" representation, make clear whether you mean low in intrinsic dimensionality ID (a nonlinear notion) or in linear dimensionality? If you mean ID, then there is a serious problem, because there is a trivial way in which neural representations furnished by neural nets which receive high-dimensional sensory inputs that are determined by a few LVs (as is the case in their experiments/simulations) have low ID. To see this, firstly note that in their (non-noisy) setting, sensory observations, o_t , are (at any time step, t) determined by and hence (nonlinear) mathematical functions of latent variables, z_t (in their case, z is the (x,y,θ) vector). So we can write $o_t = f(z_t)$, where the f is sufficiently smooth and maps (nonlinearly) from a low-dim space to a high-dim one.

Now suppose that instead of feeding these sensory inputs to an RNN that learns by PL, I simply feed o_t to a (single- or multiple layer) nonlinear feedforward net with *random and untrained* weights, and smooth (e.g. tanh) neural nonlinearities. The full vector of network responses, r_t , is then another smooth nonlinear function, g , of its inputs. Hence $r_t = g(o_t) = g(f(z_t)) =: h(z_t)$, where $h := f \circ g$ is again nonlinear but smooth. The function h thus provides (at least locally) a smooth embedding of the (low-dimensional) LV space in the (high-dimensional) neural response space. This means that even though neural responses have a nominally high dimensionality, they really traverse and sit on a nonlinear (and possibly highly curved) manifold of low intrinsic dimensionality (equal to the dimension of latent variables, z_t) embedded in the neural response space. So in this example of a completely untrained and unadapted feedforward net with random connectivity (which also likely does not lead to localized RF's), the neural representation of that network nevertheless has low intrinsic dimensionality, exactly equal to the dimensionality of LVs underlying sensory inputs. But it is clear that this is just a trivial consequence of smoothness, and points out to the fact that nonlinear intrinsic dimensionality is not the right way of assessing computationally interesting properties of neural reps (rep = representation).

As another (simpler) example, the ID of the raw sensory inputs (in their noise-free setting) is also exactly the same as the dim of LVs, even though no brain has done any computations on them! Since the sensory representation has a high nominal dimension, it moreover follows that the raw sensory inputs/representation also has a high "dimensionality gain" DG (authors' definition). So again low ID and high DG are not by themselves necessarily interesting or non-trivial. These should be pointed out.

To summarize, it seems like (as suggested in various parts of the paper, e.g. in the gaussian place field toy example discussed in "Latent and neural representation spaces") by "low dimensionality" the authors mean low intrinsic dimensionality. But for the above reasons that version of low-dimensionality is by itself a potentially problematic or trivial notion, and can results from features that have nothing to do with learning.

So is the "low" dimensionality in C1, the linear dimensionality of neural representation? But if so, then the problem is that that is inconsistent with the authors' findings: the representations learned by the RNNs actually achieve linear (PR) dimensions that are quite high and close to the nominal input dimension 20 (see e.g. Fig 4a). Related to this, in the 2nd paragraph on right column of p. 5, authors write: "Fig. 4 suggests that predictive learning forms a low-D representation (Fig. 4a)", but figure 4a is showing the linear (PR) dim which actually increases as a result of PL learning and ends up with a high, not low, value (as high as would be expected, namely close to the nominal dimensionality of sensory inputs)!

My take on what seems to be happening in the RNN, during the authors' PL experiments, is that the top PC's start coding linearly/explicitly for the latent variables. But the high value of PR suggests that in addition to these few top PC's there is nevertheless significant variance in subleading PC's, which (given $DG > 1$) are nonlinearly coding VL's (but perhaps correspond to dimensions of the manifold that are highly curved, or may linearly code the sensory observations?). If this is accurate, it would be an

interesting result, and has to be pointed out clearly.

2. Related to previous point 1, and claim C1: in the intro you write: "Our central question is whether a recurrent neural network (RNN) trained on this predictive learning task will extract representations of the underlying low-dimensional latent variables." Please define exactly what you mean by "extract representations of low-dim LV's" and provide evidence appropriate for that more specific claim (the more exact version need not appear in the Intro of course, but should appear early enough in the Results section).

For example, to be meaningful, "extract representations of ... latent variables" cannot simply mean that "neural representation/response-vector has high mutual information with LVs", because again the raw sensory inputs (in their experiments) have maximal mutual information with LV's (due to information processing inequality), which no processing by the RNN can increase. The random untrained feedforward net is again another trivial example, which naively has "uncovered the low-dimensional latent space structure".

For these well-known reasons the field has actually converged on defining notions of "explicit information" (about LVs) in a neural representation (see e.g. Hong et al. 2016 or other work by the DiCarlo lab), by using e.g. *linear* readouts or decoders (or more generally simple/limited-capacity readouts). In fact the Canonical Correlation Analysis done by the authors (which I found to be the most interesting part of the study) has a similar flavor, as it quantifies the *linear* correlation between LV's and the top PCs of the neural representation. (See comment 3 below).

So perhaps the central question should be "whether a recurrent neural network (RNN) trained on this predictive learning task will extract *explicit* representations of the underlying low-dimensional latent variables." Where you can use linear correlations/decodability for the technical definition of "explicit representation".

And then to address that modified question, various linear decoding (of LVs from RNN) analyses could perhaps also be explicitly added to the paper (in addition to the CCA analyses, which does suggest that at least x and y LV's are linearly decodable), e.g. showing that performance of linear decoders improves over training.

Other statements that can be modified/made more specific/precise along the above lines:

-The lines 15-18 of abstract, starting with "we show that..."

-lines 3-5 of "Predictive learning in a RNN", e.g. by replacing "uncovers the low-dimensional latent space structure" with "creates a linearly decodable representation of latent space" (if that is indeed true).

- 3rd line of 2nd paragraph on p. 5: "clearly represents" → "explicitly represents" (with that notion previously defined in the paper.)

3. I would (early in the result sections if not in Intro) give a more precise definition of LSST by e.g. replacing (as you say in intro) "latent space signal transfer, wherein *information* about nonlinearly encoded latent variables moves into the linearly defined top principal components of the representation as learning progresses" with something like: "latent space signal transfer, wherein as learning progresses latent variables become increasingly linearly correlated with (or linearly decodable from) the linearly defined top principal components of the neural representation."

This is indeed the notion they use (in the CCA analysis), it just needs to be clearly stated, so it doesn't evoke trivial examples like those I gave above.

Again, the point is to emphasize that linear correlation and linear decodability are key, as I explained in 2.

4. Regarding claim C3: The problem with this claim is twofold:

First, the authors have hardly provided evidence that PL *robustly* leads to localized RFs. There is some partial evidence in Supp Fig 4. But as seen there, various modifications in the PL setting e.g. not providing actions, colors or distances seem to lead to partial or strong delocalization of RFs.

Second, non-predictive learning, e.g. by feedforward nets, can also lead to localized RFs (see the work of C. Pehlevan and colleagues which the authors cite). Which means this feature cannot be simply thought of as a "signature of predictive learning".

5. I think the crucial robustness analysis shown in supplementary figure 3 and 4 needs to be extended, and those figures should be moved to the main paper (if journal format allows). For example (related to 4 above) localization of RF's should be quantified in Fig 4, and the scalar quantification can be plotted as a function of training time (as in supp. fig. 3).

6. In 2nd paragraph of right column on p.5, authors write "The transfer of latent variable information to the first PCs of the representation is tightly connected to the linear and non-linear dimensionality of the representation, as discussed in more depth in the Suppl. Mat."

However, I found the discussion in "Signal transfer analysis" weak, with possible mistakes (see my comment 13 below). If this cannot be improved, then the quoted claim has to be weakened/modified.

7. To show that C1-3 are "signatures" of PL, as authors claim throughout the manuscript, so that if one observes those features in experimental data, it provides evidence for PL in the brain, the authors have to show evidence that PL is necessary for those features (if not sufficient). However, as I pointed out above, for some of the three features PL is obviously not necessary. I pointed out localized receptive fields that do emerge in non-predictive feedforward autoencoders that have a positivity constraint (C. Pehlevan and M. Chklovskii et al).

And for claim 1, if based on the ID notion of dimensionality, there are trivial examples (of which I gave two above).

So establishing that any of these features is an exclusive signature of PL, needs more work. Otherwise the claims have to be weakened.

For example, saying that these features *can* arise from PL, or observing them in the brain *may* indicate that brain performs PL.

In "Signatures of predictive" subsection of Discussion the authors say that a high DG is a signature of PL. I argued (in my two "trivial" examples above) that the raw sensory inputs (that have not been processed by any learning brain) or untrained random feedforward nets also furnish high DG representations in the setting studied by authors (i.e. when "observations to be predicted arise from an environment with an underlying low-dimensional latent structure"). So a high DG is clearly not a *signature* of PL, and the claims here have to be corrected.

"Locality in the manifold" is a very interesting notion, and the proposal to check in experimental data is great. (But again, it hasn't been convincingly argued that that's a signature of PL and cannot commonly arise from other non-PL frameworks.)

9. Last sentence before the section "Control simulations that test...": authors state that in the case of the non-predictive learning, DG does not increase. But Figure 5f clearly shows that DG does increase in this case too —just not as much as in the predictive case. Also since convergence of learning happens faster in the non-predictive case, the x-axis limits should be modified for that case in Figs 5b, 5e and 5f, so as not to squish the interesting part in these plots and partially hide changes due to (non-predictive) learning.

10. In the section "A neural network mechanism for low-D..." the authors start by some heuristic arguments that, given the use of the linear approximation for both $\phi(x)$ (sensory representation) and g (RNN dynamics). By itself, this part would at most be considered as arguments for a linear network in a "linear" sensory environments (i.e. one that has linear latent variables). They also make conclusions about the rank (aka linear dimensionality) of R . Being aware of this, in the second paragraph of p. 7, the authors try to provide arguments for the nonlinear case ("by allowing x^* [the linearization point] to change in time") and to link the linear rank of R to the nonlinear intrinsic

dimensionality of the representation. But I found these arguments too hand-wavy, making it hard to even know if they are self-consistent. For example, what does it exactly mean "the representation r ... represents a collection of local linear maps indexed by the position of the agent in the latent space"? (Same paragraph.)

11. Next the authors try to provide an explanation for the localized RF's that emerge in their simulation. They start by mirroring the reasoning given in the work of Pehlevan and Chklovskii. However, the key ingredient there is the positivity of neural responses without which RF localization cannot be established. On the other hand, the authors point out that, in their case, even without the positivity constraint (e.g. with the tanh nonlinearity) their RNNs still develop localized RFs. This clearly points out to the explanation given not being the right/full explanation, since the explanation, unlike the phenomena, relies on positivity!

12. The calculations in first section of supp. Material have several mistakes. A first indication of that is that while PR is a dimensionless number, the result in Eq. 4 is dimensional (has units of 1/length). The correct result I believe is actually

$$PR = \text{Area}/(2\pi \sigma^2),$$

which is dimensionless (and scales as $1/\sigma^2$ and not as $1/\sigma$). Here's an outline:

First, assuming the gaussians are normalized to integrate to 1 when integrated over x , then the -1 terms in lines 2-4 of Eq 2 should really be $1/\text{Area}$ (where Area is the area of the region over which x_t is assumed to be uniformly distributed). Second, the gaussians are 2-dimensional, but it seems like the authors relied on the normalization constant for 1D gaussians. Noting the above and replacing temporal averages with spatial ones (i.e. integrate over x spatially, but normalize by dividing by Area), I find that:

$$C_{ij} = 1/(4\pi \sigma^2) \exp(-\Delta^2/(4\sigma^2)) - 1/\text{Area}$$

(there's actually another factor of $1/\text{Area}$ multiplying both terms which however is inconsequential for calculating PR). From this it follows that for N neurons:

$$\text{Tr}(C)^2 = N^2 [1/(4\pi \sigma^2) - 1/\text{Area}]^2 \approx N^2 1/(4\pi \sigma^2)^2$$

(where I assumed $\text{Area} \gg 4\pi \sigma^2$ in the approximation),

$$\text{And } \text{Tr}(C^2) \approx N^2/\text{Area}/(8\pi \sigma^2),$$

From which the result for PR follows.

The result is not only a dimensionless number, but also intuitive: $\text{Area}/(2\pi \sigma^2)$ is approximately the number of *non-overlapping* RF's fitting in the area of size Area. The linear dimensionality PR measures the number of uncorrelated/independent activity directions with significant variance. RF's that are too close and overlapping are highly correlated and redundant. So increasing RF's in the grid by increasing overlaps does not increase dimensionality. The latter is proportional to the number of non-overlapping RF's covering the area. This will generalize to RF's on higher d -dimensional latent spaces, as follows:

$$PR \sim \text{Hyper-Volume}/\sigma^d$$

which can be pointed out.

13. I found two problems with the arguments of Supp. section "Signal transfer analysis". First why should the top eigenvector (PC) necessarily be the zeroth harmonic Fourier vector? Is there really a general mathematical reason? Second, if x_0 denotes the true location, then neural activity vector in this gaussian RF model has components:

$r_i = G(x_i - x_0)$ where x_i is the center of the i -th RF. Then projection of the r vector onto the zeroth-harmonic Fourier vector n is:

$$\sum_i G(x_i - x_0)$$

which, assuming a closely-spaced and regular grid of RF's (which the authors assume too), can be approximated (up to a constant) by

the integral:

$$\int G(x - x_0) dx$$

As long as x_0 is in the middle of the grid far from its boundaries, this integral would be roughly constant and independent of x_0 (due to normalization of G), and not "not equal to x_0 if the response is normalized to one" as the authors assert.

So it looks like this section needs major corrections.

Minor comments:

- Figure 1f: how much data was used to construct these tuning curves? Provide some control? (Basically to assure the reader that any tuning seen is not simply result of "noise" due to small data.)
- in line -6 (6th from last) of left column on p. 3: replace "neuron" with "neuron's place field".

- In fig 4: I would add separate curves for CC's of x and y with different PC's.

- the quantity plotted in Fig 4c-d is the canonical correlation, not "canonical correlation analysis". So those y -labels have to be modified. Panel titles can be changed to "latent space explicit information flow" or linear information flow...

- Fig 5e: show (maybe in a supplementary figure) the result obtained by using other algorithms for ID as done in Fig. 4b (to follow the authors' own note in Methods that the recommended practice is "computing and reporting several (here, five) different estimates of ID based on distinct ideas").

- This sentence from subsection "Discovering latent structure..." in discussion sounded tautological to me: "Thus, the problem of discovering the low-dimensional, latent structure in complex, high-dimensional dynamic signals becomes that of discovering the variables that parameterize a low-dimensional neural response manifold."

Can be written better to clarify?

- Methods "Neural network model": give equations (!) to clearly describe the model.

- Methods "Description of environment": " A Gaussian filter of variance 2 is then used, for each color channel, to make the color representations smooth". What does 2 mean? In what units? (Grid space/tile?)

- Supplementary Material "Robustness of our findings": for each case (bullet point) please provide, in the text, the legend label used in Figures S3-S4 for that case.

- Supplementary Material "Emergence of localized activity": What do they exactly mean by $r_t = f(x_t)$? At any time-step the activity of the RNN is not simply a function of its current input but of the (sufficiently recent) history of its inputs. It's thus not clear what exactly they mean by $r_t = f(x_t)$ or whether that assertion is correct/self-consistent. It's true that the results (in the main part) suggest that projections of r_t onto the first PC's of the rep code directly/linearly for LV's, x_t , but this is not necessarily true of projections of r onto the next several prominent (given the high PR) PC's.

Revision: Predictive learning as a network mechanism for extracting low-dimensional latent space representations

Stefano Recanatesi^{1,*}, Matthew Farrell², Guillaume Lajoie^{3,4}, Sophie Deneve⁵, Mattia Rigotti^{6,+}, and Eric Shea-Brown^{1,2,7,+}

¹University of Washington Center for Computational Neuroscience and Swartz Center for Theoretical Neuroscience; Seattle, WA

²Department of Applied Mathematics, University of Washington; Seattle, WA

³Department of Mathematics and Statistics, Université de Montréal; Montreal, Canada

⁴Mila - Quebec Artificial Intelligence Institute; Montreal, Canada

⁵Group for Neural Theory, Ecole Normal Supérieur; Paris, France

⁶IBM Research AI; Yorktown Heights, NY

⁷Allen Institute for Brain Science; Seattle, WA

*These authors share senior authorship

+Corresponding author stefanor@uw.edu

Summary of revisions

We are grateful to the reviewers for their insight, time, and very serious evaluation of our work. We deeply appreciated the content of their comments and critiques and did our best to tackle all the issues that were raised. Thanks to this work of the reviewers (and of us in return!) the paper has substantially improved in both its results and exposition. We apologize for the long time that it took, as the depth of the comments required and deserved an extensive and careful revision including substantial new analyses.

We present a heavily improved and revised manuscript. Here is a quick summary of what has changed. This is followed by detailed point by point replies to each reviewer.

- **A clear characterization of the predictive learning framework.** The main result of our paper is that predictive learning enables the extraction of the latent space underlying sensory observations. We now provide three analyses to characterize this process: predictive error, latent signal transfer, dimensionality analysis. The first of these is new to the revised manuscript, and the previously central role of dimensionality gain has been replaced by more balanced treatment of the features of the learned neural representation, as well as new theoretical arguments more directly linked to prediction. Additionally, in the revision we do not claim localized neural activity (e.g. place cells) to be a necessary consequence of predictive learning, but rather that this phenomenon arises in most of our simulations and is in agreement with the trends highlighted by our metrics, relegating arguments as to why this may occur to the supplemental material.
- **Addition of two new benchmark tasks.** Several concerns were raised regarding the use of spatial exploration as a primary benchmark task for our results. In the revised manuscript we present results for two entirely new tasks and network settings which complement the prior results for spatial exploration. The first is a card-game task, with completely discrete

and discontinuous observations and a distinct feed forward network architecture, which is now presented in the beginning of the manuscript; the second is an arm-reaching task with a simple muscle model, now presented towards the end of the manuscript. In each, we show how predictive learning extracts the latent space underlying the task specific observations.

- **Clarified implications for hippocampal research.** Related to the use of spatial exploration, we tackled important reviewer concerns regarding the overall motivation and broader implications of our results for the hippocampal literature by carefully rewriting the manuscripts in several parts. The introduction and first section are now fully re-written, and do not point to the hippocampus in a specific way. Rather, we discuss the implications of our results for hippocampal representations in the discussion, where we also more carefully point out both the biological and modeling limitations of our framework. We thus present our findings in a more appropriately general setting throughout, and clearly state the shortcomings of our paper in modeling specific hippocampal functions. In sum, in our revision we have properly presented the paper not as a hippocampal model, but rather as a broad framework that may apply to hippocampal representations.
- **Improved exposition and theoretical analysis.** We have both revised the flow of our manuscript to clearly state our claims and lead the reader through them, and have here improved both our theoretical arguments and computational simulations. They are now both now more robust and much more clear in backing up the main claims of our paper. In response to good reviewer feedback, previous section 2 of the manuscript, an example analysis, has been moved to Suppl.Mat, and there is a new first figure which, as suggested, lays out the main ideas of our study. We have also more appropriately stated that predictive

learning provides "a mechanism" for the observed features of network representations, not necessarily being "the such mechanism", which was confusingly implied in some of our prior writing. We are grateful to the reviewers for inspiring these substantial revisions, and are confident that they have clarified the logic of exposition.

In addition to these changes to the main text, the Supplementary Material has also been extended and made more robust. Furthermore it includes new material, including more computational controls and a pilot data analysis on hippocampal data, all in accordance with the reviewer suggestions.

In what follows, we use teal color for our replies to the review.

Reviewer 1

Contributions.

The primary contribution of this work is to analyze with both RNNs and with some mathematics how predictive learning with an agent in a 2D environment gives rise to a low-dimensional latent representation of that environment, by learning $observation_t, action_t \rightarrow observation_{t+1}$. The work compares both the linear embedding of this representation and the nonlinear intrinsic dimensionality, and shows how, through learning, this representation becomes more pronounced, in the sense of occupying larger amounts of variance explained, as measured by signal location in the top PCs.

To my knowledge, the work is novel, interesting and technically correct. However, I admit, I have no idea what to make of it. My confusion stems from a number of sources: Comparisons to hippocampus (is this work meant to be about what the hippocampus does?). I acknowledge I am not an expert in this domain. My sense that the article is making possibly overly broad claims that predictive learning is THE thing required for low-d representations. My wondering what the recurrent nature of the RNN is bringing to the result. My strong opinion that the manuscript requires substantial effort in order to improve its readability to the broad audience it hopes to reach.

General answer. These are very important and valid concerns, and we addressed them in our revision according to three main categories:

- **Readability of manuscript and figures:** the flow of revised version has been improved, including inserting a new section in the beginning of the paper, clarifying the general vs specific relevance to hippocampus, and streamlining and narrowing the narrative throughout.
- **Robustness and generality of findings:** the revised version contains novel metrics and analysis, as well as application to new task and network settings (card game and arm-reaching movements).

- **Computational and dynamical role of RNN:** we have added a new example with a feed forward network, added new analysis, and trained more different types of network model On the underlying tasks.

Moreover we have made it clear that predictive learning should be accounted among other mechanisms for extracting latent space signal; we have changed text throughout the manuscript; quick examples are the revised title, and an example sentence from the the introduction: "This links predictive frameworks with existing mechanisms of extracting latent structure (1–3) and low-dimensional representations from data (4)." Overall we recognize that these changes were necessary, and believe that they have increased the quality and robustness of the paper.

Question 1. What is this paper really about and how does it relate to neuroscience? I read the position piece [22], highlighted in both the manuscript Introduction and Discussion. Its central tenet appears to be that, "hippocampus contributes by supporting memory necessary for successful navigation rather than by performing navigational computations per se", and "it is difficult to empirically associate the hippocampus with spatial computations per se, as opposed to memory for the spatial parameters and events relevant to ongoing behavior in space". The position appears to be primarily concerned with knocking down the idea that hippocampus function is solely related to integration of self motion cues, and rather is part of a relational processing system that includes navigation.

Indeed, in the manuscript discussion, the authors state that "Finally we note that the responses are reminiscent of the types of place-related activity observed in the hippocampus and entorhinal cortex, lending in particular mechanistic grounding to the recent proposal by (22) that the hippocampus builds a semantic relational network. We argue that relevant semantic relations are encoded by neural representation of low intrinsic dimensionality, and in turn these are being constructed by predictive learning to reflect the relevant latent variables in a task." What does this mean? Please expand this point further with examples. Did you create a semantic relational network? Can you point to it and to examples, with your concrete mechanistic model?

Answer 1a. We have rewritten the paper to localize these connections to the discussion, and to focus the message of the paper as described in our summary above. In the discussion, we have rewritten to clarify paragraphs including the ones the reviewer addresses here. The idea pointed out in the first of these paragraphs, that this paper shows an alternative explanations to the "integration of self motion cues" for the generation of place related information, is one of the outcome of our analysis. It is also correct to say, as expressed in your second paragraph, that this alternative explanation can be read in terms of semantic relational networks. In the new version of the paper we describe the idea of semantic relations in terms of "actions." While this identifications sheds light on some tasks (spatial navigation, novel card game example...) in some others it is less helpful, e.g. in

the case of text prediction (specifically predicting the next word). We believe that identifying semantic relations with the action operation and uncovering the role of the action in giving rise to the manifold provides a more concrete, and more mechanistic, presentation in our revision.

Question 1b. In what way is this manuscript useful to experimentalists (a useful paper need not be, I am just trying to help clarify and understand)? Is the idea that they can measure DG and then infer there is a low-dimensional representation based on predictive learning?

Answer 1b. This is a valid question, and our revision addresses it in two ways. First, we point out more clearly that learning is a sufficient mechanism for extracting predictive representations of underlying latent spaces, a conceptual point that we think is important to both experimental and theory communities. Second, while identifying features of predictive representations that are practically detectable in real data is hard, we have done our best in the revision, including specifying which of the features describe could follow from other mechanisms, and which (predictive error) gives a signature a prediction per se. In our revision, Suppl. Mat., we have also added an explicit example of real neural data analysis for a publicly available data set. This data analysis is only a 'pilot' but it does show how this ideas can be applied to neural data.

Question 2a. The statements are too strong given the results. Reading this paper, one would conclude that predictive learning is THE means by which low-dimensional latent structure is created in neural networks. However, there are a large number of examples of (low-dimensional) meaningful latent structure being learned in artificial networks. The authors point out word embeddings, for example. Please take the time to explain what is special about predictive learning. "This latent space signal transfer is another signature of predictive learning that we can exploit and track through training." Does this have to do strictly with predictive learning? Would another kind of non-predictive task also do this? More generally, the single task that is studied somewhere during the manuscript gets converted to the phrase "predictive learning". This feels like a far too general switch, as only a single task is studied here, aside from a few related controls.

An additional example of an overly strong conclusion is that predictive learning is the way to learn low-D latent structure because auto-encoding networks do not. One may merely conclude that auto-encoding is not great for learning low-dimensional latent structure. Going further, what is the motivation for the non-predictive case, e.g. $|o_t - y_t|^2$? Again, the language here feels a bit strong, in that there are many ways in which a task involves stimuli but is not predictive in nature. Perhaps "Non-predictive learning fails to extract low-D latent manifold" should be converted to "Auto-encoding RNNs fail to learn" E.g. what happens if you were to ask the network to predict distances and wall colors just outside of the set of angles that are typically measured, but from the same time step (o_t)? This task would not involve prediction through time

but might still learn an interesting representation of the space.

Another example: "Here, we investigate the hypothesis that representations with low-dimensional latent structure, reflecting such semantic organization, result from learning to predict observations about the world." I would be much more comfortable with: "Here, we investigate the hypothesis that *a means for generating* representations with low-dimensional latent structure, reflecting such semantic organization, result from learning to predict observations about the world."

Answer 2a. The reviewer is correct that predictive learning is a sufficient mechanism by which low-D latent structure is created in neural networks, but it is certainly not the only one – and that our writing and the way we introduced our metrics needed to change to make this clear.

In our revision we have accomplished this by carefully toning down claims that were too singular throughout, and by changing the way that we introduce our metrics of neural representations (which we no longer separately refer to as signatures of predictive learning). For example, the following statement at the introduction of our metrics now clarifies how our framework should read as a general way of accessing the extraction of a latent signal: "...predictive error, dimensionality gain and latent signal transfer. While the first of these is specific to predictive frameworks, the other two could be interpreted as general metrics to quantify the process of extraction of a low dimensional latent space from data." We also modified the specific sentences pointed out regarding 'non-predictive' models being 'auto-encoding' models and the second example regarding low-dimensional latent representations.

Question 2b. I am not certain I believe the generality of the results. "Our techniques require no advance knowledge of what the latent variables are, or even how many of them there are. The consequence is that both the number and identity of latent variables can be discovered by analysis of a learned neural response manifold, as studied in other settings by (43, 46-48). We introduce latent signal transfer as a viable way to uncover the relevant variables fig. 3d: as the response manifold is learned, the position of population responses along the manifold can be increasingly well predicted by the true low-dimensional latent variables, but increasingly poorly predicted by irrelevant variables."

Can you show this on an example that isn't so precisely setup? E.g. is there another, unrelated example you can try this on? Or are these very general and strong statements leaning on the mathematical section? Please explain.

Answer 2b. We addressed this issue of robustness in two ways. The first is in writing, by focusing our claims to be more precise as described above.

The second, and much more substantial, way is by providing two new examples as the reviewer requested, each contributing a new multi panel figure and section to the paper. The first is the card-game example which now opens the paper, generalizes the findings to discrete and completely discontinuous observations, and leads to new mathematical reasoning which elucidates the function of actions in building

up the representation and should be instrumental to readers in building a deeper intuition of the phenomena we describe. As a second wholly new example, we apply the framework to a predictive model of arm-reaching movements driven by a simple muscle model, later in the paper.

Question 3. What utility is given by the recurrence? In what way is the hidden state of the RNN being used? I.e. what are the dynamics? Stated yet another way, what would have been different or broken had the model been a feed-forward network? This is mentioned in the controls section, but basically asserted that feed-forward networks hinder the development of predictive representations with the key properties described above. Given the mathematical analysis in the "A neural network mechanism for low-D representation manifolds through predictive learning" section, it seems like understanding this is of central importance to the measure of the paper. For example, could not the linearization techniques around fixed points for RNNs be used to study this (Sussillo Barak, 2013)? Does the RNN make a big plane attractor? Understanding how the dynamics and representations interact seems pretty important, and likely accessible, and would help validate the mathematics derived in the manuscript.

I am guessing, though I am uncertain, that the latent space signal transfer is related to the dynamics required to perform the task. Given the control of an auto-encoder being able to learn, but not replicating the main dimensionality results, this seems plausible.

Answer 3. We read two concerns in this question: one related to RNN vs FFW models, and one related to RNN dynamics in our framework. We address these in turn.

In the revised manuscript we now start our exposition with a FFW model that also develops a predictive representation in a simpler setting. When we later introduce the RNN model, we now clearly state that need for a RNN in place of a FFW model stems from the need of integrating observations over time when this observations are partial. (A more technical motivation for this, as the reviewer doubtless knows well, would be a shift from the network modeling a Partially Observable Markov Decision Process rather than a Markov Decision Process.)

As rightly stated, the use of a RNN is also important as it introduces dynamics. We agree that approaching such a dynamical problem with the tools of Sussillo & Barak 2013 would aid the uncovering of possible plane attractors. We attempted using the toolbox but were unable to deploy it on our network by the term of this revision. We are in touch with Matt Golub, author of the toolbox, to overcome issues we faced in doing and hopefully we'll be able to provide results of the toolbox in future rounds of revision.

Question 4. The organization of the paper is very confusing. The jump from figure 1 being about an RNN and task setup, to figure 2 being about a mathematical analysis, back to figure 3 being about the RNN really confused me. I needed more guidance through the exposition. Just to confirm, is Figure 2 unrelated to the RNN trained on predicting next-step sensory input? If so, it was not clear to me until I got to the first

paragraph of the The learned neural representation manifold section. E.g. Figure 1 has an RNN in it, it seems pretty clear that Figure 2 is going to be about that.

How do you know the latent variables are a generative model for the observations? This phrasing confused me significantly. There is a traditional generative model $p(x|z)$ in this manuscript?

Answer 4. We certainly appreciate this point, and have very majorly reorganized the manuscript to address these organization and readability issues. We hope the revised version to be clearer in its flow. In particular the paper now starts with a clear, illustrative example, and it is streamlined throughout so to avoid theoretical detours.

With respect to the generative model phrase in particular, we agree that the term generative model was employed in the paper in a less traditional (and less technical) manner. With generative we meant simply there exists a function that generates observations for each combination of latent space variables. We do understand the issue, and in response we dropped the use of this term in the revised manuscript.

Moderate concerns.

Question 5. Fig 1a is your go-to figure for reader orientation. Respectfully, I think it is failing in that regard and needs work. E.g. I could not tell what part of 1a is the agent, where my eye should begin to look in the panel, and to where my eye should saccade next.

Answer 5. This is a very good point – thank you – and we have completely changed the content and flow of figure one in the revised manuscript. It now consists of a wholly new analysis and example task and network that we designed to orienting reader to the main ideas in the paper.

Question 6. I think more needs to be explained in the exposition about the nonlinear dimensionality methods.

Answer 6. We have now included two brief explanations of linear and nonlinear dimensionality methods in the main text and in the Methods.

Question 7. Other researchers have been working on RNN models that give rise to grid cells. It seems relevant to cite them (perhaps I missed it) even if the focus may be somewhat different. Cueva Wei, ICLR 2018 <https://arxiv.org/pdf/1803.07770.pdf> They find grid cells by converting from head direction and speed and predicting x,y position. Is this a task that would or would not give rise to the phenomena that you study in this manuscript?

Answer 7. We apologize for missing the reference to Cueva Wei – it fell out of our first submission in fact upon edits in which we decided not to address strongly the issue of grid cells. We are not sure whether the study by Cueva and Wei would give rise to the phenomena pointed out in this study. Our intuition is that while some of them (latent signal transfer) would possibly be there, it is not clear at least to us, what should happen to the dimensionality of the representation. In any case we have restored this reference in our revision.

Technical questions.

Question 8. If the latent space wasn't low-dimensional,

what would happen with a predictive learning network?

Answer 8. We provide two examples in response to this interesting question. The space of all images in ImageNet, considered as vectors, can be considered high dimensional. ID techniques point to a dimensionality between $ID = 35-45$ for the space of all images. If we imagine now images arranged in succession so as to apply predictive learning (maybe in the form of a movie), a dimensionality of 35-45 would still be learnable and not "too high" for recurrent neural networks. In other words whenever the latent space dimensionality is high, it is likely that the input/stimulus dimensionality is even higher. The difficult case would probably not be when the latent space dimensionality is high (and so is its representation in input space), but when the information regarding the latent space is only minimally available in the input – in other words an extreme case of partial observability.

An alternative to produce high-dimensional latent spaces could be to add noise to them, which makes them appear high-dimensional. Still RNNs have great capacity for denoising their input, and a noisy latent space would probably be denoised (cf. Fig. S9 noise RNN case).

Overall, our theoretical arguments are still valid independently of the dimensionality of the embedding space; a first example of these is the plot in Fig.1f that would clearly generalize to a lattice structure of arbitrary dimension.

Question 9. Is there noise in the network activations? I ask because it seems significant to the authors that the latent variables are encoded in the top PCs. However, I do not see why this really matters if there is no noise, as long as it can be read out correctly.

Answer 9. As the reviewer understands, the idea that top PCs encode for latent variables arises because signal directions are encoded strongly in the representation, while other aspects, or any noise, are encoded only with lower variance, not affecting top PCs. This is studied explicitly in the Suppl.Mat. In the revised manuscript we added a control with noise in the activations of the RNN, cf. Suppl.Mat. We didn't find any discrepancy with the model analyzed in the main paper.

Question 10. Should I make anything out of x,y going to pc 123 and theta going pcs 4,5? Typically, signals are mixed in the top PCs simply because top PCs explain highest variance and are otherwise unrelated to the task.

Answer 10. The specific ordering of x,y,theta on PCs 1-5 doesn't carry specific meaning, they could be mixed or sorted differently (that is, other networks could end up with a different mixing after learning). The important point is that, as the neural population representation is tuned to latent space variables, then the top principal components must be the latent variables themselves (cf. Suppl.Mat., Latent Signal Transfer section).

Question 11. "The latter necessarily implies a low dimensional representation, the same as latent space." Why is this necessary? Can it not be the case that the Wr_{t-1} is high-dimensional but the readout W_{out} reads out neural state

necessary? I.e. r can wander in directions orthogonal to the readout, thus potentially being high-dimensional, while not affecting performance.

Answer 11. In brief, we see how the highlighted sentence could generate confusion and we eliminated it from the revised manuscript.

In more detail, the idea regarding the representation being possibly high dimensional in directions orthogonal to the readout is very interesting. While a complete answer unsurprisingly goes beyond the scope of this specific paper, we reflected more on the issue and temporarily concluded the following. While in principle an alternative to the representation being low-dim (or low rank) could be to retain high rank "fluctuations" that lie exclusively in the nullspace of W_{out} ; during training and ongoing plasticity this nullspace keeps changing, so that it is difficult for any systematic variability to consistently meet this condition. This suggests why this alternative is unlikely to occur in trained networks in general (cf. (5, 6)), and does not occur in our simulations either. This is also pointed out in the main manuscript: "...the learning process enforces representations of the same decoded state to be nearly identical – which occurs in all of our simulations and is predicted by other numerical and theoretical studies (5, 7)."

Question 12. Can you decode the latent variables at all in the auto-encoding networks? If so, to what extent?

Answer 12. We added a linear decoding analysis in the Suppl.Mat. Fig.S7c as a comparison across all models. We found that the decoding of latent variables in autoencoding models was very poor compared to predictive networks.

Minor concerns.

There is a goodly amount of jargon in this paper. I encourage the authors to internally justify each term and see if they cannot reduce to a minimal set and thereby reduce the cognitive load involved in reading this piece somewhat.

Fig 1e caption: I do not know why the place cell is x,y coordinates of the latent space, isn't it the x, y coordinates of where the agent is?

Fig 1e,f caption: (one per small quadrant), I think some information is missing.

Fig 1f: Is that meant to be a space denoting the head direction angle? Please improve clarity

Fig 2e caption: PR dependence on the size of the Gaussian field, yet the figure shows DG on on the y axis A neural network mechanism for low-D representation Discussion of policies and off policy actions are a bit out of nowhere. While the methods are typically relegated to the end, it seems important to include the bits necessary during the main exposition so as not to confuse the reader.

Given that you train the RNN on random trajectories, it is probably worth spelling out what this means for the agents dynamical system $F(x)$, and what you would expect to learn in the RNN as a result.

Perhaps explain semantic relational network in the

discussion?

Canonical correlation analysis vs canonical covariation analysis

In discussion, did you mean Fig 4d?: "uncover the relevant variables fig. 3d: as"

Answer to Minor Concerns. We implemented changes in response to all these minor comments in the revised version. Thank you for pointing out all these issues.

Reviewer 2

Contributions.

This work explores the underlying mechanism of recurrent predictive learning methods in extracting semantic latent information. First, it provides approximate mathematical arguments, and second, it shows experimental evidence by studying the representations of a one-layer RNN model in a simplified random walk environment.

Strengths

1. **Originality:** Exploring the profound reasons for the effectiveness of predictive learning is a natural extension to this field, but to my knowledge has not been done before.
2. **Quality:** The qualitative and quantitative correlation analyses between the latent variables and the recurrent representations clearly demonstrate the advantage of predictive learning over non-predictive learning (auto-encoders).
3. **Significance:** Some analytical techniques in this paper about the neural representations, e.g. the latent signal transfer experiment in Fig. 3, might be of interest to practitioners who want to design new predictive learning methods.

General answer. Thank you for stating what we also see to be the importance of our work so clearly. We also recognized that your comments raise opportunities to further strengthen our manuscript, covering at least two broad areas:

- **Applicability of findings and framework:** while in this revision we do not extend framework to include an RL module or to operate in more real-world scenarios, we do take substantial new steps in that direction by providing strong evidence for the role and coding of actions in neural representations. In the original manuscript we recognize that we failed to clearly do so; thus our revision bridges more closely to RL. Second, we extend the scope of the findings by applying predictive learning to two other new tasks in the revision (a card-game, and a model of muscle-driven arm reaching movements).
- **Importance of deep learning machinery:** in the revised version we provide more analysis and controls (GRU

and LSTM networks among them) to highlight both the generality and limitations of the ideas we present.

Overall we believe that the suggested revisions have greatly improved the manuscript. While our submission was intended to address the intersection between neuroscience and machine learning, we realize that it left room for improvement in its exposition and its generality. We believe and hope this revised version resolves this while striking a much better balance between these fields.

Improvements.

Question 4. Clarity: This paper is not organized very clearly. I encourage the authors to use a paragraph to highlight the main contributions and the layout of this paper.

Answer 4. To improve clarity, we have fully modified the layout of the paper in this revision. We recognized the problems of flow in the narrative and worked to improve them considerably. We adopted your suggestion of stating the main contribution and layout of the paper in the new introduction: "Our goal is to build theoretical and data-analytic tools that explain why a predictive learning process leads to low-dimensional maps of the latent structure of the underlying tasks – and what the general signatures of such maps in neural recordings might be."

Question 5. Methodology: The authors introduce a new metric, the Dimensionality Gain (DG). However, it is not clear to me what the motivation of this new metric is. Is it the higher the better (though in Fig. 5f, predictive learning has a greater DG value, I cannot find significant difference between Fig. 5d and Fig. 5e)? Can I use it to testify the effectiveness of a new predictive learning method? I think the authors could further clarify its significance with more evidence or at least more qualitative explanations to the result.

Answer 5. Dimensionality Gain (DG) is a compact way of understanding how a low dimensional manifold can be twisted and warped so to be easily read out. To aid reading out information from the manifold, the higher DG the better. This metric is a valid metric to assess the quality of learning of different systems and compare different learning schemes to one another on the base of geometrical properties of the representation. Although it may appear quite abstract, it is a novel indicator which aids the understanding of "why" predictive learning is a powerful technique. We appreciate the point you made and overall the weight of DG in the revised manuscript has been reduced, and replaced with a more general discussion of dimension and how it relates to predictive learning. In particular, we now write: "On one hand, the representation is prompted to encode high-dimensional observations; on the other, it extracts the regularity of a low-dimensional latent space. While the high dimensionality of the observations is a global property, referring to the collection of many observations, the regularity of the latent space is induced on a local scale, as neural representations relate to their possible neighbors via the action. These demands lead the linear dimensionality PR, measuring a global property of the representation manifold,

and the non-linear dimensionality ID, measuring more local properties, to have opposite trends. This interpretation is supported by further experiments and the next example we study, that arm-reaching movements, in which the network is prompted to predict a lower dimensional observation signal. To encapsulate this phenomenon we suggest the metric of Dimensionality Gain (DG).."

Question 6. Experiments: All empirical results are based on a very simplified task. I have some concerns with this so-called navigation task.

1. Why did the authors make the agent take random walks instead of learning a navigation policy. With random walk, there are less dynamical coherences lying beneath the sequential observations. The process of predicting future observations might rely more on the actions.

Answer 6.1. This is an important point. First of all, we have changed the way we refer to the navigation task as the exploration task in the revised manuscript. In more depth, however, we note that we initially conducted numerous experiments attempting to simultaneously learn both the policy, through an RL mechanism, and the environment representation through predictive learning. We did run experiments on learning reward driven policies through RL modules based on the recurrent representation. This would allow for the learning of flexible policies based on the environmental representations. Although the results were promising, with phenomena like the emergence of the representation of rewards and spatial map modulation based on the policy, we decided not to include this direction of research in this first paper. The reason is that the results that we have presented here, limited to predictive learning of observations without any policy learning module, are already a substantial endeavor to analyze and understand. Adding an RL scheme on top add to the challenge (together with, of course, power and applicability). New papers from DeepMind (for example their work on unsupervised predictive memory (8)) shows this very clearly.

This said, in response to this and other comments, in the revised manuscript we have re-done mathematical analysis of the actions to clarify and highlight the strong and very interesting role of actions. As suggested by the reviewer, their role turns out to be critical, and in the revision we use them to clarify several aspects of predictive learning.

Question 6.2. The environment is too simple: the latent location variables (x , y) are discrete, the observations are merely 20d vectors. How about using a more complicated environment, like DeepMind Lab, with continuous latent variables and RGB frame observations.

Answer 6.2. We agree that the environment is too simple from the perspective of any modern application of machine learning, but believe there are good reasons to use this environment here. This is so that we can reasonably control and explain the effects of predictive learning in the network. Indeed, our choice of analyzing such a simple environment didn't come from a desire to avoid the simulation of more complex settings, such as DeepMind Lab environments, but was driven from a need to understanding what was actually

happening throughout learning.

In fact, we did (over the course of many months) simulate extensively a 3d environment with RGB colors. After trying several models we realized that although our network had an RNN as a bottleneck, it was quite important how the visual information was packed to the network. Convolutional layers for the encoding and decoding of visual information were strongly affecting learning and the environment being bigger and more complicated was not always trainable with a vanilla RNN. LSTM units for example were explored but their dynamics appeared, at least to us, distinct in several ways. Although we succeeded in training networks on predictive tasks overall, trying to comprehensively analyze representations in a network that was parsing an environment similar to DeepMind Lab arena was humbling to say the least, and we return to the simpler environment presented in the paper. Following this paper, now that we have quite a deeper understanding of the possible ways predictive representations can form and be shaped, we are starting again to analyze predictive representation in deeper and more realistic contexts. For future work we plan on comparing our current results with different predictive learning schemes – for example David Cox's error prediction framework. We hope that these efforts of ourselves and others in the community will be aided by the interpretability we were able to achieve here.

Question 6.3. The sequential action inputs might reveal the latent variable, theta, which makes the correlation analysis regarding theta and neural representations less convincing.

Answer 6.3. Yes, we do agree that the analysis on theta in our scheme is less convincing that the one of x - y : as you point out partial information regarding the angle is carried by the action. A caveat is that, in our setup, we made sure that the action wouldn't carry full information regarding the angle, as actions were discrete, providing only summary information about theta direction that can vary within an angle of 45° . We added this in the text through the discussion of Fig.3b: "Similarly the axis of symmetry for the angle θ is located closer to $\Delta t = 1$, although in this case the analysis is confounded by the fact that actions carry partial information regarding θ ."

Question 6.4. In such an easy environment, future observations are easy to predict given the very first observation and a trajectory of actions. As the training epoch grows, the neural network might have remembered these information and make prediction without any glimpse of any new observations. Thus, I have to cast doubt on the correlation analysis between the observations and the neural representation PCs.

Answer 6.4. This an insightful question, but it is not the case that, in our setup, future observations are decodable given the initial observation and a sequence of actions. The reason is that the agent's angle is a continuous variable and for each angle a distinct observation follows: Far from the wall even just a rotation of 4-6 degrees in the angle would generate a completely different set of observations. The action, however, is discrete, being the projection of the angle

onto the 8 possible 45 degrees sectors of the full 360 degrees. Given a sequence of actions it is therefore impossible to infer the agent's angle with sufficient precision to infer the observations. We have clarified this point even more in the revised Methods.

Question 6.5. Last but not least, the authors use an extremely simplified network, a one-layer RNN. I understand for mathematical analysis, simplification and approximation is acceptable. But in the experimental analysis, the authors might investigate some more frequently-used network structures, such as LSTM and GRU, which obviously violate the linear assumption in Eq. (7) and (8).

Answer 6.5. We agree that our network is highly simplified, and as the reviewer rightly pointed out it is for the sake of interpretability. This said, in our revision we now inserted as controls LSTM and GRU units as suggested, adding these results to the Suppl.Mat. Figs. s7-10. Overall our results appear to be robust to these and quite a number of variations in the model.

Question 6.6. References: The authors may cite some recent references in spatiotemporal data forecasting and model-based planning, e.g. World Model, which have successfully applied predictive learning to its downstream tasks.

Answer 6.6. Thank you for pointing this out. We were not aware of this important thread in the literature and we added citations it in our revised paper.

Question 6.7 Typos: right column of page 2: it should be Fig. 1c and Fig. 1d.

Answer 6.7. These typos have been corrected.

Conclusion.

I believe this paper is tackling an important problem, but makes some claims without strong evidence (see comment 6). I thus cannot recommend acceptance at the current time but might improve my score if enough evidence is provided.

Final remarks. We strongly appreciated the thrust of the reviewer comments above, and believe that we have provided enough new evidence in our revision to warrant its reconsideration. We hope that the reply above is helpful in resolving concerns surrounding the relationship between action and angle, as well as overall organization, LSTM/GRU units, and motivation for the predictive learning setup based on observations. Additional new evidence that strengthens our results in the revision is summarized in the below:

- Generalization of predictive learning framework to two more tasks (card game and arm-reaching movement), Figs. 1 and 6.
- Novel theoretical analysis for the joint role of actions and observations in shaping the representation.
- Systematization of metrics for tracking predictive learning and low-d representations signatures, Fig.3.
- Several new controls and analysis for the spatial navigation case as well as pilot data analysis on hippocampal data.

Reviewer 3

General answer. Reviewer 3 raises both important technical points, and concerns that are understandably focused on the connection of our study to the experimental literature – specifically concerning the relationship of our results with the hippocampal system. These legitimate concerns are appreciated and they have led to substantial revision and improvement of the manuscript. We give a careful point-by-point reply below, but first offer a summary of our reply, in three parts:

- **Hippocampus.** We agree that the submitted paper had a stronger emphasis on the hippocampus than was appropriate given the general nature of our analysis and simulations, and have worked hard on the revised manuscript to make sure that our findings wouldn't read as an hippocampal but rather a more general neural network model with some implications for the hippocampus. We added two novel tasks and networks, reorganized the flow of the paper, and fully rewrote the introduction, to motivate our work in a broad way.
- **Experimental connections.** We answered the concern regarding the way our study informs future analysis on hippocampus by adding a short Pilot Analysis on hippocampal data (9, 10) that exemplifies how our study can inform neural data analysis. While this is only a pilot as a full data analysis would require a devoted study and probably a more dedicated dataset as well, cf. Suppl.Mat. Fig. S13, we believe that it is useful in demonstrating the applicability of our results more directly.
- **Biological realism.** As suggested we softened our claims on place cell localized activities and other hippocampus related findings. In the revision we clearly state that our model doesn't aim at matching biological circuits directly, (e.g., second paragraph of the revised Discussion). Rather, our work explores the hypothesis that the brain may exploit predictive learning to build maps of varied tasks, through the extraction of the underlying latent space.

Majors.

Question 1. The manuscript does not attempt to tie the theoretical simulations and arguments back to data or established models of data (e.g. place cells, head direction cells, grid cells, boundary cells etc). I recognize that it attempts to make a connection between the latent space representation to resemble place cells but falls short of a detailed analysis and linking that to reported data.

Answer 1. This is helpful feedback – we recognize that our submission didn't tie computational results back to data well enough. In the revised version we included a pilot data analysis of publicly available hippocampal data (9, 10). We believe that this pilot analysis, included in the Suppl.Mat. Fig. S13, will serve as a useful example for future and more complete studies. We also note there

that hippocampal data recorded across long timescales (11) will be particularly promising. In our revision we also have improved our presentation of the characteristics of predictive learning in neural representations, and introduced a new feature specific to prediction, that of predictive error (see new Figs. 3a-b) which we also check successfully in the pilot data analysis. Overall, while this was admittedly not clear in our submission, we view our principal contribution as shedding light on the principles by which prediction can shape neural representations, and thanks to reviewer feedback we much more clearly describe the nature of our contribution throughout the revision and especially in its fully reworked introduction.

Question 2. The work overall and sections in detail (see examples in minors) requires better motivation. Currently, it reads: Lets take the latest training methods (RNN) and simulate a predictive task and study the latent representations in principle component space. Its unclear to me why this approach is novel or relevant to the audience of this journal. It's unclear to me how the current finding would provide a tool to analyze existing or new data in a new light. Maybe, it would help if the authors propose such a tool and take a few data sets (on place cells, head direction cells, grid cells) that have been collected in navigation tasks and apply their machinery there. E.g. the Moser lab provided data here <https://www.ntnu.edu/kavli/research/grid-cell-data>.

Answer 2. We understood that the principal objectives of our paper should have been more clearly laid out in our submission, specifically in its introduction and first main section, and welcomed the chance to improve on this in our revision. In this revision it is much more clear that, while the impact of our work relies mostly on computational and theoretical results, it goes well beyond the simple application of recent techniques to the prediction task. It is, at least to the best of our knowledge, one of the first works that asks "how should predictive representations be encoded in neural data?". Both reviewers 2 and 4 rightly describe this as:

- "Exploring the profound reasons for the effectiveness of predictive learning is a natural extension to this field, but to my knowledge has not been done before."

- "The manuscript addresses a set of key questions in theoretical neuroscience, related to emergence of useful neural representations as a result of unsupervised learning and adaptation to the sensory environment."

In our revision, we more carefully describe these contributions. We also propose and demonstrate a set of features that, taken together, characterize the implications of predictive learning for neural data.

Question 3. The task of predicting the environmental layout for the next step is rather arbitrary and constructed. It is in the line of work of auto-encoders (here with some predictive time dynamics) that exists for a long time. Tasks for navigation if this is what the authors are aiming for; which citations to work of memory representations in rats seems to suggest – are semi-supervised in nature. The current proposal has a conceptual mismatch here or implicitly assumes that such predictive auto-encoders are a pre-requisite to successful

navigation.

Answer 3. We realized that our manuscript was focused too much on the hippocampus without pointing out the strong limitations and assumptions of our model. We considered this issue very carefully during the revision process and did our best to disentangle our proposed framework from the hippocampal case, and the task of navigation in particular. In our revision we cast our findings in a more appropriately general light, in particular adding analyses and simulations of two new tasks that show the broad applicability of the predictive learning framework.

Although we still include a detailed analysis of the spatial navigation task, we have completely re-named it as random exploration; this helps make clear the limitations that the reviewer points out. Overall, in the revision we emphasize that this model has too many limitations to be considered a realistic model of the hippocampal computation, stating in the discussion: "Our model falls short in explaining mechanistically key elements of spatial maps individuated in hippocampal recordings, such as the emergence of place cells and their relation to direction or grid cells.". We further eliminated explicit reference to the hippocampus in the introduction. We hope this careful re-writing of the paper addresses concerns related to the specific applicability of our results to the case of rodents navigation.

Our current theoretical proposal is in line with others that see the emergence of relational networks from unsupervised predictive learning (12). Our goal here is to show that it is possible for a brain area, such as the hippocampus, to build spatial (latent variable) maps via predictive learning rules implemented in neural networks.

Question 4. The methods section and theoretical arguments provide the reader with some insight into the framework but the manuscript falls short in providing a reproducible description of the model. Ive also given the supplemental material a quick read and could not find a complete description there either. I acknowledge that the authors have provided source code but I suggest that this does not replace a description of the model that would allow the reader to reproduce their findings.

Answer 4. In response we enriched, in the revised manuscript, both the methods section and the supplementary material. We believe that the revised version, via these further details, has improved significantly in its clarity. More recently we received feedback from a computational neuroscience group which was able to replicate our results from the version of the paper we posted on biorxiv. Taken together we are thus confident that the revised paper now is sufficient for scientific reproducibility.

Minors (details).

Question 5. Page 2: "demonstrate our main result: that the network uncovers the low-dimensional latent space structure in the course of optimizing its future predictions..". The approach is not novel (e.g. Beardsley & Viana, Computational modeling of optic flow selectivity in MSTd, Comput Neural Sys 1998; study latent representation hidden layer in non-recurrent network to identify representations

related to optic flow patterns for a navigation task, or Olshausen & Field, Emergence of simple-cell receptive field properties by learning a sparse code for natural images. Nature 1996; study an auto-encoder for images with a sparseness constraint where representations show Gabor filter-like shapes).

Answer 5. Thank you for the references related to optic flow selectivity and sparse coding. We were not aware of the first reference and its related literature. We added these references to the revised manuscript in the discussion section. Although the approach of this literature is similar, to our judgement it deals with auto-encoders and not predictive coding or recurrent dynamics. The difference, as pointed out in our paper, is important: an autoencoder is not capable of exploiting time-dependencies in data as a predictive coding framework.

Question 6. Page 2: "The agent we consider is equipped with simple sensors that span a visual cone of 90° centered on its current direction." Why were these values chosen? Why do we have color? Does it matter? Why not use odors or whiskers as sensory input? Please be reasonable and motivate well. I'm unsure how reasonable is the RGB color space for rats; if you indeed try to simulate rat vision.

Answer 6. We acknowledge that our framework does not provide a biologically realistic account of rodent vision. The details of the sensory-like observations that we use in the special exploration task are driven by considerations closer to our computational and theoretical objectives. Specifically, our choices are driven by the necessity of having a simple model that still singles out the contribution of distance, visual anchor points (color) and actions. In this case we traded off some biological realism for simplicity and interpretability in order to distill the core results on the ability of predictive learning to extract latent space information.

This said, we have performed a number of controls to confirm the robustness of our basic results to modeling assumptions. For the visual cone we selected an angle of 90° but our simulations show that a smaller or bigger angle wouldn't change. In the Suppl.Mat. we provide a number of other controls for understanding how color vs distance information impact the results. For example we show that while color would matter (coded RGB or in other ways), knowledge of the distance could be left out while reproducing most of the results.

Question 7. Page 2: "Actions are performed by the agent with respect to its allocentric framework, so that there are nine possible choices" What's transforming the sensory signals into allocentric coordinates?

Answer 7. We did try both allocentric and egocentric frameworks. The differences between the two, in our eyes, rely on whether or not the animal has the ability to perform its actions with respect to anchor points in the environment. We are aware of a number of studies that focus on the specific transformation from egocentric to allocentric, and we do implicitly assume that such a system is in place. Our choice of the allocentric has the advantage of simplicity and interpretability in the resulting model. As highlighted by

Fig.1 in the revised manuscript, allocentric actions give a key for interpreting learning of the representation that egocentric ones would not provide. An allied consideration is that, by using precise egocentric actions, the animal would have been able to integrate its actions to build a representation of its current state; the uncertainty we insert related to allocentric actions doesn't allow this (cf. discussion above with reviewer 2).

Question 8. Page 2: "In predictive learning, the RNN learns to predict the upcoming sensory observation" Which species has a mechanism like that: To predict the shape/layout after each single step? Mental rotation tasks of 3d objects might come closest to this. But humans are poor at this task.

Answer 8. The prediction can be thought as the expectation for the observation upon taking a selected or arbitrary action. If we think on the scale of, say 100ms, then this kind of prediction could be intrinsically generated by a specific brain area simply by the learning rule used to process the inputs. There are studies showing that even spike-timing dependent plasticity could potentially lead to such representations (13–15). If the prediction, or more simply an expectation of upcoming sensory input, is exploited by local learning rules to build predictions, then multiple areas in the brain could achieve this. Also, if we consider that what could be minimized is the prediction error (similarly to the framework of Cox et al. (16)) then predictive learning could be a local circuit mechanism to build long range expectations of future events or observations that don't match the internal expectation. Most species display surprise and/or a learning behavior (e.g. conditional paradigms) when their expectation doesn't match their observations. This is a broader way of interpreting predictive learning, but it is still consistent with the framework we analyze, which is driven by predictive error-based learning in neural networks. Our study focuses on why such predictive mechanisms are effective in building representations of the latent spaces underlying the sensory information; in other words answering "how do maps of different tasks arise from predictive learning rules?".

Question 9. Page 3: "How does the neural population as a whole represent the latent space?" Why does this question matter? Please motivate.

Answer 9. To make sure that the importance of this question is understood we expanded the question in the revised manuscript and introduced it from a different angle: "The metrics just introduced capture properties of the neural representation at the population level via useful numbers that can be plotted over the course of learning. Here, we pause to visualize the underlying population representations in two complementary ways. The first visualization is directly related to the metric of latent space signal transfer..". As the reviewer likely knows, much modeling work now focuses on the coherent activity of the population of neurons encoding for specific variables rather than the separate response profiles of individual neurons. The right way to mathematically tackle how population of neurons encode for variables is to ask geometric, statistical or information theoretic questions of the kind we posit above: "how does the

population specifically code for these variables of interest?" Here the "how" stands for a deeper mathematical meaning which can be described in terms of physical observables such as the ones we highlight here. Signal to noise ratio is an often used metric that stands as an example, but many others are needed to explain "how" a neural population codes for a given variable.

Question 10. Page 3: We can view the tuning curve of a single neuron (Fig. 2b) on the response manifold to obtain the manifold tuning curve of this neuron (Fig. 2d). In the next section we will analyze in more depth. The projection is neighborhood preserving when visualized in a 3D PC1-3 space. Do you see this as a result of visualizing in PCs or as part of the mechanism or as part of the task being fairly continuous?

Answer 10. The smoothness of receptive fields on the response manifold can be induced by the task being fairly continuous. However, their locality, in terms of the cells having mostly a single bump response profile, cannot: smoothness could still induce cells with multiple bumps or alternate profiles on the response manifold. The fact that cells have single bumps suggests that they might be tiling the response manifolds, acting as place-manifold cells. This is an important observation because it points to the possibility that place fields are generated through place-manifold cells rather than the opposite. In the modeling community there is a rising interest in how this could occur through unsupervised learning algorithms (e.g. (17)).

Question 11. Page 3: "The fact that the representation manifold has two dimensions is revealed by a measure known as Intrinsic Dimensionality (ID), whose formal definition relies on concepts of Riemannian geometry for smooth manifolds" A motivation for why we need to do this analysis is missing. What's the intuition behind this analysis?

Answer 11. In our revision we more explicitly state the motivation for studying intrinsic dimensionality. This was introduced in the supplementary theoretical section S2.2 where we present the idea that linear dimensionality, often used in terms of number of PCs necessary to describe 90% of neural activity variability, falls short of capturing the real dimensionality of the manifold. Most studies report a PC like dimensionality but this often used metric can be very high, pointing to a high dimensionality, even when the dimensionality is low. It may be helpful to think of a sheet of paper crumpled in a 10 dimensional space. The Intrinsic Dimensionality of it would be 2 - it is still a sheet of paper - but assessing dimensionality by means of PCs would often lead to a much higher number depending on the space in which the paper was crumpled - 10 in this case. This can be misleading, giving to a very poor estimate of the number of sensory or behavioral variables encoded by the neural population. Intrinsic dimensionality is then a different measure of dimensionality that should complement the usual metrics already used. This theoretical discussion has been revised but moved to the Suppl.Mat., a more intuitive explanation has been introduced in the main text on page 6 paragraph "Dimensionality Gain".

Question 12. Page 4: "This shows that the agent's location x,y is systematically encoded in the first three PCs, while PCs four and five encode the agent's orientation , Fig. 3b." Interesting. Do you think this is a result of choosing PC1-5 in that way? There seems to be some form of neighboring preserving smoothness here; but this time it is split across modalities.

Answer 12. We do confirm that in our findings neighboring preserving smoothness, often described as local similarity preservation, appears to be present. The specific order of PC1-5 could vary (e.g. having the orientation represented on PCs 3-4 instead of PCs 4-5) but the coding properties would be the same and this could arise with the correlation structure across modalities. For example if orientations are on average decorrelated from locations then it makes sense that their representations would also be orthogonal occupying different PCs. It could still be smooth across all variables but would develop in different dimensions. This is in line with recent findings in the group of M. Chklovsky where it was shown that local similarity preservation is a key concept underlying manifold place-cells (17). The idea is that manifold place-cells or place-cells in general are the outcome of algorithms which encode information by preserving only local similarities but not global similarities: the relative distance of neighboring points is preserved in the representation, but not that of distant points. These are important emerging ideas regarding representation and coding in neural circuits. We present a lengthier explanation of this in the theoretical sections of the Suppl.Mat. Sec. S2.5

Question 13. Page 8: "We find that predictive learning in recurrent neural networks (RNNs) leads to an intriguing answer, as it automatically constructs a low dimensional neural representation of the latent space." Here predictive seems to be the key. Could you show the same holds true for other tasks, e.g. for a grasping, visual tracking, ...?

Answer 13. Thank you for this suggestion. We inserted two new tasks in the revised manuscript: a card-game and a arm-reaching task following your suggestion to study grasping. Your suggestion of modeling grasping, and thus motor cortex activity, definitely aids in establishing our results and generalize our framework. In these tasks the principles and implications of predictive learning generalize directly.

Question 14. Page 9: "Finally we note that the responses are reminiscent of the types of place-related activity observed in the hippocampus and entorhinal cortex, lending in particular mechanistic grounding to the recent proposal" Can you quantify the place cells more strictly, e.g. by fitting Gaussian models and reporting r^2 or by defining a place cell score?

Answer 14. We defined a place cell score based on a glm poisson fitting. in the Suppl.Mat. Fig. S13, Pilot Data Analysis section, we now show a summary statistics of place fields statistics for both our simulations and hippocampal data.

Final Remarks. In addition to important technical concerns, we recognized that most of the comments regarding the implications of our framework derived from the fact that

our paper chose as benchmark a spatial navigation task strictly related to hippocampal coding properties, as well as confusingly specific ways in which we introduced our main ideas in the introduction. This "strong" focus on hippocampal activity was not our singular interest – rather our interest was to show properties of predictive learning and its correlates in neural populations. We agree that a level of biological realism and behavioral detail to match the standards of a model of hippocampal spatial maps are clearly missing from our study, as you carefully pointed out. To resolve this in our revised manuscript we clearly state our contributions beginning from the introduction, where the hippocampus is not even mentioned. Furthermore we added two new substantial task examples of predictive coding, following your very good suggestion regarding motor driven movements. These changes led to significant overall improvement in the manuscript.

Reviewer 4

The manuscript addresses a set of key questions in theoretical neuroscience, related to emergence of useful neural representations as a result of unsupervised learning and adaptation to the sensory environment. Specifically the authors study whether recurrent networks trained via predictive learning to predict their future sensory inputs, will learn to uncover and represent the (typically low-dimensional) latent variables underlying their sensory inputs. Addressing such questions is very welcome, and the current study could potentially be very impactful and influential.

However, I find several major shortcomings in the current manuscript. My two major criticisms are:

- I found some of the main claims and proposed hypotheses of the paper not to be clearly stated or sufficiently elaborated to distinguish them from trivial (or in some cases tautological) interpretations. In other cases I found the evidence provided for the claims to be incomplete.

- The mathematical analysis provided to explain some of the findings was quite lacking in rigor.

I understand that a fully rigorous analysis may be difficult or impossible. But at least highly non-rigorous steps of the arguments could/should e.g. be corroborated by clearly presented directly relevant empirical evidence from the simulations.

Below I will go into the above (and other more minor comments) in more detail. But first, I'll summarize the three main claims of the paper (all summarized at the end of Introduction):

Predictive Learning (PL) by an recurrent neural net (RNN) creates or leads to:

C1. low-dimensional neural representations, reflecting the few latent variables (LVs) underlying sensory observations.

C2. what they call Latent Space Signal Transfer (LSST).

C3. localized receptive fields (RFs).

Moreover C1-3 are taken as signatures of PL.

General Answer. First of all thank you for your careful critique, which was precise and uncovered mistakes for which we apologize. The revised manuscript has been strengthened in several aspects. Those specific to your comments include:

- we significantly modified and improved our theoretical analysis in several points. Specifically we now account for the role of actions in building the representation in both an explicit intuitive way and a more abstract one, the latter relating to group actions.
- we carefully changed our statements and lexicon to be more precise following several of your suggestions.
- we clarified several crucial points regarding our claims regarding dimensionality and metrics. This includes replacing prior rank-based analysis and explaining a more nuanced role for ID metrics.

Question 1. Regarding C1: Be more precise in stating this hypothesis and potentially modify it. In particular, when you say "low-dimensional representation, make clear whether you mean low in intrinsic dimensionality ID (a nonlinear notion) or in linear dimensionality? If you mean ID, then there is a serious problem, because there is a trivial way in which neural representations furnished by neural nets which receive high-dimensional sensory inputs that are determined by a few LVs (as is the case in their experiments/simulations) have low ID. To see this, firstly note that in their (non-noisy) setting, sensory observations, o_t , are (at any time step, t) determined by and hence (nonlinear) mathematical functions of latent variables, z_t (in their case, z is the (x,y,θ) vector). So we can write $o_t = f(z_t)$, where the f is sufficiently smooth and maps (nonlinearly) from a low-dim space to a high-dim one.

Now suppose that instead of feeding these sensory inputs to an RNN that learns by PL, I simply feed o_t to a (single- or multiple layer) nonlinear feedforward net with *random and untrained* weights, and smooth (e.g. \tanh) neural nonlinearities. The full vector of network responses, r_t , is then another smooth nonlinear function, g , of its inputs. Hence $r_t = g(o_t) = g(f(z_t)) =: h(z_t)$, where $h := f \circ g$ is again nonlinear but smooth. The function h thus provides (at least locally) a smooth embedding of the (low-dimensional) LV space in the (high-dimensional) neural response space. This means that even though neural responses have a nominally high dimensionality, they really traverse and sit on a nonlinear (and possibly highly curved) manifold of low intrinsic dimensionality (equal to the dimension of latent variables, z_t) embedded in the neural response space. So in this example of a completely untrained and unadapted feedforward net with random connectivity (which also likely does not lead to localized RF's), the neural representation of that network nevertheless has low intrinsic dimensionality, exactly equal to the dimensionality of LVs underlying sensory inputs. But it is clear that this is just a trivial consequence of smoothness, and points out to the fact that nonlinear intrinsic dimensionality is not the right way

of assessing computationally interesting properties of neural reps (rep = representation).

As another (simpler) example, the ID of the raw sensory inputs (in their noise-free setting) is also exactly the same as the dim of LVs, even though no brain has done any computations on them! Since the sensory representation has a high nominal dimension, it moreover follows that the raw sensory inputs/representation also has a high *dimensionality gain* DG (authors definition). So again low ID and high DG are not by themselves necessarily interesting or non-trivial. These should be pointed out.

To summarize, it seems like (as suggested in various parts of the paper, e.g. in the gaussian place field toy example discussed in "Latent and neural representation spaces") by "low dimensionality the authors mean low intrinsic dimensionality. But for the above reasons that version of low-dimensionality is by itself a potentially problematic or trivial notion, and can results from features that have nothing to do with learning.

So is the *low* dimensionality in C1, the linear dimensionality of neural representation? But if so, then the problem is that that is inconsistent with the authors' findings: the representations learned by the RNNs actually achieve linear (PR) dimensions that are quite high and close to the nominal input dimension 20 (see e.g. Fig 4a). Related to this, in the 2nd paragraph on right column of p. 5, authors write: "Fig. 4 suggests that predictive learning forms a low-D representation (Fig. 4a), but figure 4a is showing the linear (PR) dim which actually increases as a result of PL learning and ends up with a high, not low, value (as high as would be expected, namely close to the nominal dimensionality of sensory inputs)!"

My take on what seems to be happening in the RNN, during the authors' PL experiments, is that the top PCs start coding linearly/explicitly for the latent variables. But the high value of PR suggests that in addition to these few top PC's there is nevertheless significant variance in subleading PCs, which (given $DG > 1$) are nonlinearly coding VLs (but perhaps correspond to dimensions of the manifold that are highly curved, or may linearly code the sensory observations?). If this is accurate, it would be an interesting result, and has to be pointed out clearly.

Answer 1. There are several layers to your important concerns here. We offer a reply in two parts, first on the "triviality of low-ID". Second, we'll reply regarding the different components that make up the representation, thus LVs, actions and observations.

Your mathematical observation regarding the fact that any smooth nonlinear function of, say, 2 LVs would have $ID = 2$ is very important, and is something we considered very carefully: it contributed to us modifying the principal theoretical arguments in our paper to be based on the role of actions, and the way that we describe below.

Here, we first wish to address: is our result trivial or tautological? From mathematical considerations regarding smoothness of maps it indeed appears so, but there are rich and often counter-intuitive effects connected to multi scaling

and ID. As one example, say we input to a feedforward untrained network a full movie. A movie is mathematically speaking parametrized by a single variable t – time – and for simplicity can be considered to be smooth in all its components. So the ID of the representation of a movie in any layer of a deep network with 'smooth' convolutional or ffw layers should be one, at least according to an argument based on parameterization such as that given the above. Still we know that there is a lot of complexity in the frames of a movie. Indeed as soon as we subsample the movie and shuffle frames, then we may not be able to put the frames back together in the right order based on similarity, which suggests that frames that are far apart in the temporal parametrization are similar. In other words a movie can in principle form a folded trajectory that gets closer to itself at multiple scales. The consequence is that if we look at a nearly infinitesimal scale we would see a 1-dimensional manifold, but as soon as we zoom out a bit (e.g., subsample the frames) we may find ourselves in a, say, 50 dimensional space.

As the reviewer may know, this problem is known as multiscaling and it points out the fact that at different scales the same manifold may be characterized by different dimensions. Importantly, different measures of ID do actually measure the dimensionality in different ways and at different scales. So when we point out in the text that the dimensionality of the extracted manifold decreases through learning and that different measures agree with each other (Fig. 3f) we are also saying, in a sense, that its effective dimensionality becomes more regular and "smoother" across all scales. If we compare the ID of the representation in the non predictive case (Fig. 4e) we see that it is higher. This again doesn't mean that there is no low (in this case two) dimensional parametrization of the manifold, but rather that its statistical properties appear higher dimensional.

The reviewer's ideas and criticisms along these lines let us to realize that this is a more important aspect of our study than we realized initially, and that it needs to be treated much more clearly in the paper. We have worked very hard on this and believe that our extensive revision resolves in several ways. For starters, in the main manuscript in the section describing Fig. 4e we now state: "Importantly one could expect that ID would be equal in the case of predictive and non-predictive coding on the ground that the observations could locally be considered a 3d manifold and that the network performs smooth transformations. This argument fails because ID, despite being a local measure, looks at statistical properties of the manifold points in a statistical way (cf. Methods). So if the manifold appears higher dimensional, despite having a parametrization which is low-dimensional, like in the case of our observations then ID would point to a higher dimension. In other terms ID is sensitive to the manifold's smoothness and can be taken as a measure of it for manifolds known to be parametrized by an equal number of variables. This problem is known in the literature as multiscaling and different ID measures are more or less robust to it (18)."

We have also added new analyses that directly address the underlying point. The first of these is an entirely new task

and network setting that now is the focus of a new figure one and fully new first main section of the paper. This is the card game task. Here, the dimensionality of the representation is low and parameterized by discrete latent variables, while the observations are completely uncorrelated, fully independent functions of the latent variables with no smoothness, thus being high-d. The second of these is new theoretical analysis which has replaced the arguments given before. This new analysis relies on the effect of actions on network representation, but does not employ the same smoothness based dimensionality arguments which legitimately lead to the reviewer's concerns in our submitted manuscript. We have to repeat that we are grateful for the reviewer's observations and critique, as these changes have greatly improved our arguments and the strength of our paper. To the reviewer's second point regarding other possible components encoded in the representation, it is indeed clear that this is occurring. The CCA analysis can be interpreted as an analysis of decoding throughout stages of learning. To complement this, we also added a regression analysis for the latent variables (Fig. 3b. Results show that observations are coded into the representation with implications, as you pointed out, on higher PCs coding for observations (e.g. as Fig. 5b shows).

Question 2. Related to previous point 1, and claim C1: in the intro you write: "Our central question is whether a recurrent neural network (RNN) trained on this predictive learning task will extract representations of the underlying low-dimensional latent variables. Please define exactly what you mean by "extract representations of low-dim LVs and provide evidence appropriate for that more specific claim (the more exact version need not appear in the Intro of course, but should appear early enough in the Results section). For example, to be meaningful, extract representations of & latent variables" cannot simply mean that neural representation/response-vector has high mutual information with LVs, because again the raw sensory inputs (in their experiments) have maximal mutual information with LVs (due to information processing inequality), which no processing by the RNN can increase. The random untrained feedforward net is again another trivial example, which naively has "uncovered the low-dimensional latent space structure".

For these well-known reasons the field has actually converged on defining notions of "explicit information (about LVs) in a neural representation (see e.g. Hong et al. 2016 or other work by the DiCarlo lab), by using e.g. *linear* readouts or decoders (or more generally simple/limited-capacity readouts). In fact the Canonical Correlation Analysis done by the authors (which I found to be the most interesting part of the study) has a similar flavor, as it quantifies the *linear* correlation between LVs and the top PCs of the neural representation. (See comment 3 below).

So perhaps the central question should be "whether a recurrent neural network (RNN) trained on this predictive learning task will extract *explicit* representations of the

underlying low-dimensional latent variables. Where you can use linear correlations/decodability for the technical definition of "explicit representation". And then to address that modified question, various linear decoding (of LVs from RNN) analyses could perhaps also be explicitly added to the paper (in addition to the CCA analyses, which does suggest that at least x and y LVs are linearly decodable), e.g. showing that performance of linear decoders improves over training.

Other statements that can be modified/made more specific/precise along the above lines:

-The lines 15-18 of abstract, starting with "we show that."

-lines 3-5 of "Predictive learning in a RNN", e.g. by replacing "uncovers the low-dimensional latent space structure" with "creates a linearly decodable representation of latent space" (if that is indeed true).

- 3rd line of 2nd paragraph on p. 5: "clearly represents" – > "explicitly represents" (with that notion previously defined in the paper.)

Answer 2. We implemented these suggestions in the revised manuscript. Specifically we adopted the notion of "explicit representation" as you suggested using linear correlations/decodability as a way to probe whether explicit representations are present or not. This has been done throughout the manuscript. We believe this to have critically improved the manuscript.

Question 3. I would (early in the result sections if not in Intro) give a more precise definition of LSST by e.g. replacing (as you say in intro) "latent space signal transfer, wherein *information* about nonlinearly encoded latent variables moves into the linearly defined top principal components of the representation as learning progresses with something like: "latent space signal transfer, wherein as learning progresses latent variables become increasingly linearly correlated with (or linearly decodable from) the linearly defined top principal components of the neural representation. This is indeed the notion they use (in the CCA analysis), it just needs to be clearly stated, so it doesn't evoke trivial examples like those I gave above. Again, the point is to emphasize that linear correlation and linear decodability are key, as I explained in 2.

Answer 3. We factored out the word information from the paper as you suggested. We agree it could lead to confusion – instead we rewrote relevant sentences in terms of linear encoding of variables. We also adopted the phrasing you suggested in the introduction.

Question 4. Regarding claim C3: The problem with this claim is twofold: First, the authors have hardly provided evidence that PL *robustly* leads to localized RFs. There is some partial evidence in Supp Fig 4. But as seen there, various modifications in the PL setting e.g. not providing actions, colors or distances seem to lead to partial or strong delocalization of RFs. Second, non-predictive learning, e.g. by feedforward nets, can also lead to localized RFs (see the work of C. Pehlevan and colleagues which the authors cite). Which means this feature cannot be simply thought of as a signature of predictive learning.

Answer 4. These are important points. In the revised version of the paper the role of actions is much more clear, including their possible role in structuring the representation and manifold cells. Our claim regarding RFs has also been revised; in fact, now it doesn't read as a strong claim (like C3) but as a way of thinking about the emergence of the phenomena highlighted with connections (and as the reviewer points out, still some distances) to and from the current literature.

The revised Fig. 3 in the manuscript details the three metrics we use to characterize predictive learning. As we don't claim predictive learning to be a mechanism for RFs, we simply show evidence that such representations can arise through predictive learning, which is, we believe, of interest. In other words we seek to relay an observation rather than a mechanistic explanation of how RFs arise from PL. To clarify this, we state explicitly the neural RFs don't have to be localized due to dimensionality or other constraints, but in our simulations we do consistently find this locality. Thus, in the revised manuscript, we point to the cited papers of C. Pehlevan and colleagues, pointing out the differences but still highlighting the potential connections via the way networks may exploit and integrate local vs global similarities in its processing.

Question 5. I think the crucial robustness analysis shown in supplementary figure 3 and 4 needs to be extended, and those figures should be moved to the main paper (if journal format allows). For example (related to 4 above) localization of RFs should be quantified in Fig 4, and the scalar quantification can be plotted as a function of training time (as in supp. fig. 3).

Answer 5. The quantification of RFs localization has now been introduced as an analysis in Fig. S13. Unfortunately the length of a Nat.Comm. paper doesn't allow for integrating these results into the main manuscript – but we did expand our treatment with more controls and analysis. This said we didn't include plots of the sparsity metric throughout learning, because quantifying the size of the "bumps" of the representation was more challenging than we expected and the metrics we tried were not as robust as we would have liked. In more detail, many of the place-fields developed in the network are elongated and poorly captured by gaussian fits, at the same time other metrics like L1 or L2 norms failed to be sufficiently robust.

Question 6. In the 2nd paragraph, right column on p.5, the authors write "The transfer of latent variable information to the first PCs of the representation is tightly connected to the linear and non-linear dimensionality of the representation, as discussed in more depth in the Suppl. Mat. However, I found the discussion in "Signal transfer analysis" weak, with possible mistakes (see my comment 13 below). If this cannot be improved, then the quoted claim has to be weakened/modified.

Answer 6. The theoretical argument has been substantially improved as explained in answer to question 13.

Question 7. To show that C1-3 are *signatures* of PL, as authors claim throughout the manuscript, so that if one

observes those features in experimental data, it provides evidence for PL in the brain, the authors have to show evidence that PL is necessary for those features (if not sufficient). However, as I pointed out above, for some of the three features PL is obviously not necessary. I pointed out localized receptive fields that do emerge in non-predictive feedforward autoencoders that have a positivity constraint (C. Pehlevan and M. Chklovskii et al). And for claim 1, if based on the ID notion of dimensionality, there are trivial examples (of which I gave two above). So establishing that any of these features is an exclusive signature of PL, needs more work. Otherwise the claims have to be weakened. For example, saying that these features *can* arise from PL, or observing them in the brain *may* indicate that brain performs PL.

Answer 7. In the revised manuscript we modified the structure of our claims (for example C3 is not a claim anymore as explained above) and provide a clearer understanding of PL and its characteristics. We have changed writing throughout our paper to make it clear that do not claim that predictive learning is the "only" mechanism through which the features of neural representations at hand – dimensionality and signal transfer – may arise. In addition, in our revision we substantially strengthen theoretical arguments to show why it is one, perhaps surprisingly simple, mechanism by which they arise robustly. We have also added a new metric, of Predictive Error, which complements the others and which together with them forms a more unique signature of predictive learning.

Question 8. In "Signatures of predictive" subsection of Discussion the authors say that a high DG is a signature of PL. I argued (in my two "trivial" examples above) that the raw sensory inputs (that have not been processed by any learning brain) or untrained random feedforward nets also furnish high DG representations in the setting studied by authors (i.e. when "observations to be predicted arise from an environment with an underlying low-dimensional latent structure). So a high DG is clearly not a *signature* of PL, and the claims here have to be corrected.

Answer 8. We modified this section of the discussion. As addressed in previous comments, we do find that low-d representations that reflect the latent space are still a meaningful feature arising from PL. At the same time, we agree that DG is a more nuanced measure to track and we have rewritten text to drop the stronger claims on DG here. To this point we also state clearly that the trends highlighted by DG and latent signal transfer are not specific to predictive learning, but could be taken as general signatures of manifold extraction: "... these are predictive error, dimensionality gain and latent signal transfer. While the first of these is specific to predictive frameworks, the other two could be interpreted as general metrics to quantify the process of extraction of a low dimensional latent space from data..."

Question 9. "Locality in the manifold" is a very interesting notion, and the proposal to check in experimental data is great. (But again, it hasn't been convincingly argued that that's a signature of PL and cannot commonly arise from other non-PL frameworks.)

Answer 9. The fact that locality could arise from non-PL frameworks is now stated clearly in the main paper. This is stated along with the fact that localized manifolds have not been shown to be a necessary consequence of PL – we now express clearly that PL is not an exclusive mechanism for locality but rather a "possible" one. As we write above, we think this is important not because it shows that PL can result in localized RFs, but rather because it shows that localized RFs could simply be the outcome of several different learning rules. In the revised manuscript we also add a pilot analysis of experimental data.

Question 10. Last sentence before the section "Control simulations that test": authors state that in the case of the non-predictive learning, DG does not increase. But Figure 5f clearly shows that DG does increase in this case too just not as much as in the predictive case. Also since convergence of learning happens faster in the non-predictive case, the x-axis limits should be modified for that case in Figs 5b, 5e and 5f, so as not to squish the interesting part in these plots and partially hide changes due to (non-predictive) learning.

Answer 10. We now inserted in the discussion of this figure the following sentence: "In both the predictive (blue line) and non-predictive scenario (red line) DG grows through learning but in the former case the increase is more pronounced." This change is in addition to overall reducing the role of DG in the revised manuscript. We considered the modification of the figure suggested and matched the overall average learning between the predictive and non-predictive case "stretching" the learning axis to aid the figure's readability.

Question 11. In the section "A neural network mechanism for low-D the authors start by some heuristic arguments that, given the use of the linear approximation for both $\phi(x)$ (sensory representation) and g (RNN dynamics). By itself, this part would at most be considered as arguments for a linear network in a linear" sensory environments (i.e. one that has linear latent variables). They also make conclusions about the rank (aka linear dimensionality) of R . Being aware of this, in the second paragraph of p. 7, the authors try to provide arguments for the nonlinear case ("by allowing \hat{x} [the linearization point] to change in time") and to link the linear rank of R to the nonlinear intrinsic dimensionality of the representation. But I found these arguments too hand-wavy, making it hard to even know if they are self-consistent. For example, what does it exactly mean "the representation r ... represents a collection of local linear maps indexed by the position of the agent in the latent space"? (Same paragraph.)

Answer 11. In the revised manuscript we provide new and distinct theoretical arguments which replace the former rank-based arguments about which the reviewer raises justified concerns. We believe that these address the primary concerns while explicitly pointing out the role of actions and updates in forming network representations. Addition to direct new arguments for the role of actions in building representations we also establish a link between their role and that of group representations. We believe that the revised text is not just more specific to prediction and more accurate but more likely to inspire future studies that will extend the

analysis and make it more rigorous.

Question 12. Next the authors try to provide an explanation for the localized RFs that emerge in their simulation. They start by mirroring the reasoning given in the work of Pehlevan and Chklovskii. However, the key ingredient there is the positivity of neural responses without which RF localization cannot be established. On the other hand, the authors point out that, in their case, even without the positivity constraint (e.g. with the tanh nonlinearity) their RNNs still develop localized RFs. This clearly points out to the explanation given not being the right/full explanation, since the explanation, unlike the phenomena, relies on positivity!

Answer 12. Both before and after our submission, we were aware of all the constraints regarding the framework of localized non-negative matrix factorization, having carefully studied the work of Pehlevan and Chklovskii and even having based a journal club and rotation student's project on these extremely interesting papers! Thus while we, more clearly in our revision, do not claim that that framework matches perfectly ours (and we hope to establish a more direct link between the two in separate future work), we do believe that the Chklovskii framework exposed in (17) is still the best way, at least for the time being, to understand why local similarities give rise to local-manifold fields. The fact that in our simulations this result extends to representations that are not necessarily positive could be due to the fact that this phenomena is more general than the mathematical scenario in which it was first defined by the original work of Pehlevan-Chklovskii, a prospect which while unresolved we find quite exciting. We are very clear about this in the revised manuscript, cf. Suppl.Mat. Sec. S2.5 that is now accompanied by a new figure comparing positive and non-positive activation functions.

Question 13. The calculations in first section of supp. Material have several mistakes. A first indication of that is that while PR is a dimensionless number, the result in Eq. 4 is dimensional (has units of 1/length). The correct result I believe is actually $PR = Area/(2ps^2)$, which is dimensionless (and scales as $1/s^2$ and not as $1/s$). Here's an outline: First, assuming the gaussians are normalized to integrate to 1 when integrated over x , then the -1 terms in lines 2-4 of Eq 2 should really be $1/Area$ (where Area is the area of the region over which x_t is assumed to be uniformly distributed). Second, the gaussians are 2-dimensional, but it seems like the authors relied on the normalization constant for 1D gaussians. Noting the above and replacing temporal averages with spatial ones (i.e. integrate over x spatially, but normalize by dividing by Area), I find that: $C_{ij} = 1/(4ps^2) \exp(-D^2/(4s^2)) - 1/Area$ (there's actually another factor of $1/Area$ multiplying both terms which however is inconsequential for calculating PR). From this it follows that for N neurons: $Tr(C)^2 = N^2[1/(4ps^2) - 1/Area]^2 \approx N^2 1/(4ps^2)^2$ (where I assumed $Area \gg 4ps^2$ in the approximation), And $Tr(C^2) \approx N^2/Area/(8ps^2)$, From which the result for PR follows.

The result is not only a dimensionless number, but also intuitive: $Area/(2ps^2)$ is approximately the number of

non-overlapping RF's fitting in the area of size Area. The linear dimensionality PR measures the number of uncorrelated/independent activity directions with significant variance. RF's that are too close and overlapping are highly correlated and redundant. So increasing RF's in the grid by increasing overlaps does not increase dimensionality. The latter is proportional to the number of non-overlapping RF's covering the area. This will generalize to RF's on higher d-dimensional latent spaces, as follows: $PR\ Hyper - Volume/s^d$

Answer 13. We are grateful for your precision and care in working out the calculation – you are absolutely correct here. We corrected the original calculation in the revised manuscript pointing out the case of higher dimensional receptive fields as well. In the original calculation it was indeed negligence on our part not to consider the area; the entire argument now reads much better.

Question 14. I found two problems with the arguments of Supp. section "Signal transfer analysis". First why should the top eigenvector (PC) necessarily be the zeroth harmonic Fourier vector? Is there really a general mathematical reason? Second, if x_0 denotes the true location, then neural activity vector in this gaussian RF model has components: $r_i = G(x_i - x_0)$ where x_i is the center of the i-th RF. Then projection of the r vector onto the zeroth-harmonic Fourier vector n is: $\sum_i G(x_i - x_0)$ which, assuming a closely-spaced and regular grid of RFs (which the authors assume too), can be approximated (up to a constant) by the integral: $\int G(x - x_0)dx$ As long as x_0 is in the middle of the grid far from its boundaries, this integral would be roughly constant and independent of x_0 (due to normalization of G), and not "not equal to x0 if the response is normalized to one" as the authors assert. So it looks like this section needs major corrections.

Answer 14. This precise comment pointed out, correctly once again, an error which we have (easily in this case) corrected, as well as an opportunity for further interesting additions to the paper. There are two questions raised here related to the analysis – and is much appreciated. To the first, regarding a general mathematical reason for the zeroth harmonic being the top eigenvector, we note that this follows it as a result for circulant and Toeplitz matrices alike. For circulant matrices, as you probably know, the eigenvector space is well established in terms of the harmonic modes and the eigenvalues can also be computed in terms of the matrix "band" properties. For Toeplitz matrices, the case we analyze, the results are under development in the linear algebra community currently but what is known now is sufficient to support our analysis. In a few words, the results on circulant matrices extend to Toeplitz matrices, but boundary conditions and finite size effects in toeplitz matrices influence the way the results extend to this case. For example in the Toeplitz scenario some of the discrete harmonics, that are eigenvectors of a circulant matrix, may not be eigenvectors of the Toeplitz case simply because of their value at the borders of the matrix. We explain some of these details in the Suppl.Mat. pointing to the right

references.

To your second point, the vector that carries information regarding the space is not the constant vector (zeroth fourier component), but the first fourier component. This was wrongly stated in our previous manuscript (thank you again for catching this) and now carefully explained in the same section of Suppl.Mat. This component correlates with the latent variable very strongly. We revised this section it is now much improved.

Minor comments: .

- Figure 1f: how much data was used to construct these tuning curves? Provide some control? (Basically to assure the reader that any tuning seen is not simply result of noise due to small data.)
- in line -6 (6th from last) of left column on p. 3: replace "neuron" with "neuron's place field".
- In fig 4: I would add separate curves for CC's of x and y with different PC's.
- the quantity plotted in Fig 4c-d is the canonical correlation, not "canonical correlation analysis". So those y-labels have to be modified. Panel titles can be changed to "latent space explicit information flow" or linear information flow.
- Fig 5e: show (maybe in a supplementary figure) the result obtained by using other algorithms for ID as done in Fig. 4b (to follow the authors' own note in Methods that the recommended practice is "computing and reporting several (here, five) different estimates of ID based on distinct ideas").
- This sentence from subsection "Discovering latent structure" in discussion sounded tautological to me: "Thus, the problem of discovering the low-dimensional, latent structure in complex, high-dimensional dynamic signals becomes that of discovering the variables that parameterize a low-dimensional neural response manifold. Can be written better to clarify?"
- Methods "Neural network model": give equations (!) to clearly describe the model.
- Methods Description of environment: "A Gaussian filter of variance 2 is then used, for each color channel, to make the color representations smooth". What does 2 mean? In what units? (Grid space/tile?) Answer. Yes in space tile space.
- Supplementary Material Robustness of our findings: for each case (bullet point) please provide, in the text, the legend label used in Figures S3-S4 for that case.

Answer to Minor Comments. We have addressed all this points in the revised manuscript, resolving the underlying problems. Thank you again for the very careful and accurate reading.

Question 15. -Supplementary Material "Emergence of localized activity": What do they exactly mean by $r_t = f(x_t)$? At any time-step the activity of the RNN is not simply a function of its current input but of the (sufficiently recent) history of its inputs. It's thus not clear what exactly they mean by $r_t = f(x_t)$ or whether that assertion is correct/self-consistent. It's true that A the results (in the main part) suggest that projections of r_t onto the first PCs of the rep code directly/linearly for LVs, x_t , but this is not necessarily

true of projections of r onto the next several prominent (given the high PR) PCs.

Question 15. We have clarified the assumptions made for this argument and its limitations in the revised version.

Bibliography

1. David M. Blei. Build, compute, critique, repeat: Data analysis with latent variable models. *Annual Review of Statistics and Its Application*, 1:203–232, 2014. Publisher: Annual Reviews.
2. Ruslan Salakhutdinov. Learning deep generative models. *Annual Review of Statistics and Its Application*, 2:361–385, 2015. Publisher: Annual Reviews.
3. Bomin Kim, Kevin H. Lee, Lingzhou Xue, and Xiaoyue Niu. A review of dynamic network models with latent variables. *Statistics surveys*, 12:105, 2018. Publisher: NIH Public Access.
4. Laurens Van Der Maaten, Eric Postma, and Jaap Van den Herik. Dimensionality reduction: a comparative. *J Mach Learn Res*, 10:66–71, 2009.
5. Stefano Recanatesi, Matthew Farrell, Madhu Advani, Timothy Moore, Guillaume Lajoie, and Eric Shea-Brown. Dimensionality compression and expansion in deep neural networks. *arXiv preprint arXiv:1906.00443*, 2019.
6. Matthew Farrell, Stefano Recanatesi, Guillaume Lajoie, and Eric Shea-Brown. Recurrent neural networks learn robust representations by dynamically balancing compression and expansion. September 2019.
7. Matthew Farrell, Stefano Recanatesi, Guillaume Lajoie, and Eric Shea-Brown. Recurrent neural networks learn robust representations by dynamically balancing compression and expansion. September 2019.
8. Greg Wayne, Chia-Chun Hung, David Amos, Mehdi Mirza, Arun Ahuja, Agnieszka Grabska-Barwinska, Jack Rae, Piotr Mirowski, Joel Z. Leibo, Adam Santoro, Mevlana Gemic, Malcolm Reynolds, Tim Harley, Josh Abramson, Shakir Mohamed, Danilo Rezende, David Saxton, Adam Cain, Chloe Hillier, David Silver, Koray Kavukcuoglu, Matt Botvinick, Demis Hassabis, and Timothy Lillicrap. Unsupervised Predictive Memory in a Goal-Directed Agent. *arXiv:1803.10760 [cs, stat]*, March 2018. arXiv: 1803.10760.
9. Eva Pastalkova, Vladimir Itskov, Asohan Amarasingham, and György Buzsáki. Internally generated cell assembly sequences in the rat hippocampus. *Science*, 321(5894): 1322–1327, 2008.
10. E Pastalkova, Y Wang, K Mizuseki, and G Buzsáki. Simultaneous extracellular recordings from left and right hippocampal areas ca1 and right entorhinal cortex from a rat performing a left/right alternation task and other behaviors. *CRCNS*, 2015.
11. Yaniv Ziv, Laurie D. Burns, Eric D. Cocker, Elizabeth O. Hamel, Kunal K. Ghosh, Lacey J. Kitch, Abbas El Gamal, and Mark J. Schnitzer. Long-term dynamics of CA1 hippocampal place codes. *Nature Neuroscience*. 16(3):264–266, March 2013. ISSN 1097-6256. doi: 10.1038/nn.3329.
12. Kimberly L. Stachenfeld, Matthew M. Botvinick, and Samuel J. Gershman. The hippocampus as a predictive map. *Nature Neuroscience*, 20(11):1643–1653, November 2017. ISSN 1546-1726. doi: 10.1038/nn.4650.
13. Rajesh P. N. Rao and Terrence J. Sejnowski. *Predictive Coding, Cortical Feedback, and Spike-Timing Dependent Plasticity*. 2002.
14. Pau Vilimelis Aceituno, Masud Ehsani, and Jürgen Jost. Synaptic time-dependent plasticity leads to efficient coding of predictions. *arXiv preprint arXiv:1907.10879*, 2019.
15. Robert Urbanczik and Walter Senn. Learning by the dendritic prediction of somatic spiking. *Neuron*, 81(3):521–528, 2014. Publisher: Elsevier.
16. William Lotter, Gabriel Kreiman, and David Cox. Deep predictive coding networks for video prediction and unsupervised learning. *arXiv preprint arXiv:1605.08104*, 2016.
17. Anirvan Sengupta, Mariano Tepper, Cengiz Pehlevan, Alexander Genkin, and Dmitri Chklovskii. Manifold-tiling localized receptive fields are optimal in similarity-preserving neural networks. *bioRxiv*, 2018. doi: 10.1101/338947.
18. Francesco Camastra and Antonino Staiano. Intrinsic dimension estimation: Advances and open problems. *Information Sciences*, 328:26–41, January 2016. ISSN 0020-0255. doi: 10.1016/j.ins.2015.08.029.

Reviewers' Comments:

Reviewer #1:

Remarks to the Author:

The authors have done an admirable job of revising the manuscript. They have answered all of my concerns, in particular eliminating both my primary concern of an overly broad interpretation and my secondary concern of lack of readability.

In my opinion the manuscript is ready for publication.

Reviewer #3:

Remarks to the Author:

Thanks for replying to all my concerns and improving the manuscript.

Outstanding issues

Major Q3: The authors fall short in showing that these representations matter. They build upon the assumption --in the field-- that these representations improve or are necessary to perform in learning tasks. To resolve this the authors provide several metrics (in Fig 3) that show that representations for predictive learning have "better" information measures than representations for non-predictive learning. Note, that these information theoretic metrics are rather abstract and are not easily followed by the general reader.

In my view there are several ways this can be addressed:

a) Show that the performance on the original tasks (e.g. successful grasping, successful navigation, successful plays) improves when using these representations compared to e.g. random representations while including an ergonomic constraint to make the approaches comparable.

b) Clearly map all representations in the latent space back to physiological data where possible and available. Thanks for taking this direction for place cells.

c) Clearly state in the introduction that your work is based on the assumption that successful task solving requires continuous, maybe ergonomically efficient representations -- which you think your algorithm can discover for a variety of tasks.

Minor Q8: Thanks for your reply. I'm still skeptical that humans learn by predicting, then comparing the predicted to reality, and then adjusting their predictions over time toward reality. If you could provide some behavioral results that would be great. I was thinking about mental rotation tasks -- but humans do not too well on that.

New Minor

Fig 4 is missing legends for the colors -- neither could I find the information in the figure caption.

Revision: Predictive learning as a network mechanism for extracting low-dimensional latent space representations

Stefano Recanatani^{1,*}, Matthew Farrell², Guillaume Lajoie^{3,4}, Sophie Deneve⁵, Mattia Rigotti^{6,+}, and Eric Shea-Brown^{1,2,7,+}

¹University of Washington Center for Computational Neuroscience and Swartz Center for Theoretical Neuroscience; Seattle, WA

²Department of Applied Mathematics, University of Washington; Seattle, WA

³Department of Mathematics and Statistics, Université de Montréal; Montreal, Canada

⁴Mila - Quebec Artificial Intelligence Institute; Montreal, Canada

⁵Group for Neural Theory, Ecole Normal Supérieure; Paris, France

⁶IBM Research AI; Yorktown Heights, NY

⁷Allen Institute for Brain Science; Seattle, WA

*These authors share senior authorship

+Corresponding author stefanor@uw.edu

Indications

In what follows, we use teal color for our replies to the reviewers while we use blue color for new or edited parts in the manuscript.

Reviewer 1

Remarks to the Author.

The authors have done an admirable job of revising the manuscript. They have answered all of my concerns, in particular eliminating both my primary concern of an overly broad interpretation and my secondary concern of lack of readability.

In my opinion the manuscript is ready for publication.

Answer. Thank you for the critical feedback and suggestions in the previous round, which played a major role in improving both the content and exposition of the manuscript.

Reviewer 3

Remarks to the author.

Thanks for replying to all my concerns and improving the manuscript.

Answer. The reviewer is very welcome, and we believe that we are able to do the same with the outstanding issues below.

Outstanding issues.

Question 1) The authors fall short in showing that these representations matter. They build upon the assumption –in the field– that these representations improve or are necessary to perform in learning tasks. To resolve this the authors provide several metrics (in Fig 3) that show that representations for predictive learning have "better" information measures than representations for non-predictive learning. Note, that these information theoretic metrics are rather abstract and are not easily followed by the general reader.

In my view there are several ways this can be addressed:

- Show that the performance on the original tasks (e.g. successful grasping, successful navigation, successful plays) improves when using these representations compared to e.g. random representations while including an ergonomic constraint to make the approaches comparable.
- Clearly map all representations in the latent space back to physiological data where possible and available. Thanks for taking this direction for place cells.
- Clearly state in the introduction that your work is based on the assumption that successful task solving requires continuous, maybe ergonomically efficient representations – which you think your algorithm can discover for a variety of tasks.

General answer 1. To address the reviewer's outstanding issues we made changes and/or additions to the paper that address each of their suggestions (a-c): we added clarifying text for point a), performed a new analysis on newly included electrophysiology data from motor cortex for point b), leading to a new supplementary section and new Fig. S14, and added an additional paragraph and text to the manuscript as suggested for point c). This is further detailed below.

Answer 1a. We agree with the reviewer about the importance of demonstrating the improvement in the original tasks that can be achieved with predictive representations. Fortunately – although we would agree with the reader that we failed to clearly point out its relevance in the main text – we believe that this has already been addressed in the case of our central (navigation) task, in the Supplementary Material. We have therefore revised the main text to add the following new sentence explicitly mentioning the advantage of predictive models over non-predictive ones, together with a pointer to where this is developed: "These results show that predictive models outperform non-predictive models in the encoding of latent variables, at least when such encoding is probed by means of linear measures (cf. Fig. S7)".

In more detail, readers are now pointed clearly to the Supplementary Material sections which address how the performance on the navigation task improves when using predictive representations compared to other learning algorithms. Fig. S7a-c compares a number of models on the basis of the representation of space they generate (Figs. S8-10). In this context "successful learning" is interpreted as the one that builds a highly accessible representation of space, meaning that the agent's position can be decoded linearly. Thus Fig.S7c shows that predictive learning schemes outperform non-predictive schemes as they generate representations where space information is more linearly decodable.

Specifically, we based a new analysis (new Suppl. Mat. Fig. S14) on openly available recording data from primate primary motor cortex for a task which, while not identical to the reaching task simulated in the paper, shares important similarities: the monkey controlled an on-screen cursor and was rewarded for moving that cursor to an indicated reach target, with multiple targets presented per trial (1, 2). In particular, both action and latent spaces are readily identifiable. In the new analysis of Suppl. Mat. Fig. S14 we consider a possible latent space to be the position of the cursor and show neural tuning curves on both this latent space and the PC space of neural activities. The results closely parallel the existing analysis, mentioned here by the reviewer, of the Hippocampal data in Fig. S13, and we believe that this parallel provides insightful and appropriate addition to the paper.

Answer 1c. Thank you for stressing this point. We have now added a sentence clearly stating this among the assumptions listed in the introduction. The added text, readable in blue in the updated draft, is: "In this paper, motivated by the literature suggesting that these efficient representations are instrumental for the brain's ability to solve a variety of tasks (8–10), we ask: How does such an organization of information emerge?"

Question 2) (Q8). Thanks for your reply. I'm still skeptical that humans learn by predicting, then comparing the predicted to reality, and then adjusting their predictions over time toward reality. If you could provide some behavioral results that would be great. I was thinking about mental rotation tasks – but humans do not too well on that.

Answer 2). We now state the importance of analyzing human behavior, and the fact that it is still unknown whether humans exploit predictive learning, among the open-questions of our discussion. We do this by highlighting a potential link between predictive learning and Reinforcement Learning that ought to be explored: "Furthermore, it will be important to develop a formal connection between predictive learning mechanisms (73, 74) and Reinforcement Learning (RL) paradigms (9, 75) in both model-free and model-based schemes (76–78). This has the potential to build a general framework that could uncover predictive learning behavior in both animals and humans. One step here would be to extend existing RL paradigms to scenarios where making predictions is important even in the absence of rewards

(79–82)."

Beside these links, we deem it plausible that such mechanisms may be important for human behavior. The phenomenon that the reviewer described as "predicting, then comparing the predicted to reality" can be thought of as computing a "state prediction error", which is a type of signal predicted by model-based reinforcement learning and has been recently been observed in humans in fMRI experiments (see e.g. Gläscher et al. *Neuron* 66.4 (2010): 585-595). More generally, the error of an expectation is naturally related to the idea of surprise and, to a certain extent, novelty, which are known to correlate with memorability of experienced objects or events (3, 4). In other words, prediction errors are signals that are known to be computed by the brain and have been proposed to drive learning. How such learning signals are mechanistically implemented in the brain, and how closely their algorithmic description is actually captured by our predictive learning framework, we believe to be interesting matters for future investigation.

Beside normative behavioral theories like reinforcement learning, there is a rich literature regarding our ability to copy behaviors of others (copying behavior). This can also be broadly interpreted in a predictive framework as copying someone's behavior boils down to having a template for a specific behavior and adjusting our actions (movements etc) to match this template at each moment in time, i.e. taking at each instant the action which produces the expected future behavior. In this context actions are learned till the behavioral template is matched. This is a critical way of learning especially for children (5, 6).

While the roles of prediction will remain under investigation and debate, we believe that these points and references are sufficient to reflect the plausible importance of predictive learning, and the verifiable geometrical features of the neural representations that it predicts, across species.

Minor comments.

Question 3. Fig 4 is missing legends for the colors – neither could I find the information in the figure caption.

Answer 3. Thank you for pointing this out. We fixed this oversight, which was due to colors not being rendered properly because of a figure conversion error. Other figures that were affected by the same issue to a lesser extent were also fixed.

Bibliography

1. Patrick N Lawlor, Matthew G Perich, Lee E Miller, and Konrad P Kording. Linear-nonlinear-time-warp-poisson models of neural activity. *Journal of computational neuroscience*, 45(3):173–191, 2018.
2. Matthew G Perich, Patrick N Lawlor, Konrad P Kording, and Lee E Miller. Extracellular neural recordings from macaque primary and dorsal premotor motor cortex during a sequential reaching task. *CNRS.org*, 2018. doi: <http://dx.doi.org/10.6080/K0FT8J72>.
3. Andrew Barto, Marco Mirolli, and Gianluca Baldassarre. Novelty or surprise? *Frontiers in psychology*, 4:907, 2013.
4. Lisa K Fazio and Elizabeth J Marsh. Surprising feedback improves later memory. *Psychonomic Bulletin & Review*, 16(1):88–92, 2009.
5. Mark Nielsen. Copying actions and copying outcomes: social learning through the second year. *Developmental psychology*, 42(3):555, 2006.
6. Harriet Over and Malinda Carpenter. Putting the social into social learning: Explaining both selectivity and fidelity in children's copying behavior. *Journal of Comparative Psychology*, 126(2):182, 2012.

Reviewers' Comments:

Reviewer #3:

Remarks to the Author:

Thanks for addressing my outstanding concerns in the rebuttal and manuscript.